# On Weights and Quotas for Weighted Majority Voting Games

**Xavier Molinero** [1,*] **, Maria Serna** [2] **and Marc Taberner-Ortiz** [3]

1  Mathematics Department, Universitat Politècnica de Catalunya · BarcelonaTech, E-08222 Terrassa, Spain
2  Computer Science Department, Institut de Matemàtiques de la UPC-BarcelonaTech (IMTech),
   Universitat Politècnica de Catalunya · BarcelonaTech, E-08034 Barcelona, Spain; mjserna@cs.upc.edu
3  School of Mathematics and Statistics, Universitat Politècnica de Catalunya · BarcelonaTech,
   E-08034 Barcelona, Spain; marctaberner@gmail.com
*  Correspondence: xavier.molinero@upc.edu

**Abstract:** In this paper, we analyze the frequency distributions of weights and quotas in weighted majority voting games (WMVG) up to eight players. We also show different procedures that allow us to obtain some minimum or minimum sum representations of WMVG, for any desired number of players, starting from a minimum or minimum sum representation. We also provide closed formulas for the number of WMVG with $n$ players having a minimum representation with quota up to three, and some subclasses of this family of games. Finally, we complement these results with some upper bounds related to weights and quotas.

**Keywords:** weighted majority voting games; experiments; minimum (sum) representation; canonical representation; quotas and weights

## 1. Introduction

Simple games are the simplest model to study decision systems in which the yes/no has to be decided cooperatively. A simple game is described by a monotone set of winning coalitions, i.e., the subsets of participants that can force a yes decision on an issue. One of the most natural human ways to reach a decision is through voting. Weighted majority voting games (WMVG) conform to the most widely studied subclass of simple games. In a weighted majority voting game each player has associated a weight and, for a coalition to win, it is required that the cumulative weight of the coalition will be equal to or larger than a determined quota. Weighted majority voting games were defined in 1944 by von Neumann and Morgenstern [1], but similar ideas were used one year before by McCulloch and Pitts [2] to define the *Threshold Logic Unit*, the first artificial neuron. Some years later, they were deeply studied in the 50s in the context of simple game theory [3]. Since then weighted majority voting games have also been studied in many different contexts under different names (see for example [4–11]). Various political and economic organizations use weighted voting games for structural or constitutional purposes. For example, the United Nations Security Council, the Electoral College of the United States, the International Monetary Fund, or the European Union [12–15]. Voting power is also relevant in joint stock companies where each shareholder gets votes in proportion to the ownership of a stock [16] and in political and financial decision making [17]. Several applications in decision theory of voting systems have been done over stochastic modelling Szajowski and Yasuda [18], Noroizifari et al. [19], safety critical systems Singamsetty and Panchumarthy [20] or intrusion detection Moukafih et al. [21], among others.

Simple games can also be described by monotone Boolean formulas. Therefore, the problem of enumerating the set of simple games is the same as the well known Dedekind problem of determining the number of distinct monotone functions of $n$ variables. Although Dedekind first considered this question in 1897, no satisfactory answer is yet known. Dedekind's numbers are known only for values of $n \leq 8$ and also an upper bound of $10^{42}$

for $n = 9$ is known [22]. In consequence, many attempts have been made to enumerate (or to count) subfamilies of simple games (see for example [22–25]). Weighted majority voting games up to 6 players were enumerated in [26,27]. Enumerations/countings for 8 voters can be found for example in [28–31] and for 9 voters in [32]. These counting results for weighted majority voting games (up to isomorphism) are given in Table 1. Note that, in the values provided in Table 1, isomorphic games are counted only once. The two trivial games, the one in which no coalition wins or the one in which all coalitions (including the empty set) win, are not counted.

**Table 1.** Number of WMVG, WMVG without dummies (https://oeis.org/A000619 (accessed on 18 October 2021)) [33], Non Seft-dual WMVG without dummies, and Self-dual WMVG without dummies (https://oeis.org/A003184 (accessed on 18 October 2021)) up to isomorphism.

| # Players | #WMVG | #WMVG w/o Dummies | # Non Self-Dual WMVG w/o Dummies | # Self-Dual WMVG w/o Dummies |
|---|---|---|---|---|
| 1 | 1 | 1 | 0 | 1 |
| 2 | 3 | 2 | 2 | 0 |
| 3 | 8 | 5 | 4 | 1 |
| 4 | 25 | 17 | 16 | 1 |
| 5 | 117 | 92 | 88 | 4 |
| 6 | 1111 | 994 | 980 | 14 |
| 7 | 29,373 | 28,262 | 28,148 | 114 |
| 8 | 2,730,164 | 2,700,791 | 2,698,456 | 2335 |
| 9 | 993,061,482 | 990,331,318 | 990,158,360 | 172,958 |

In the enumeration of weighted majority voting games it is usual to count the number of games without dummies or/and duality. Recall that due to monotonicity, we can represent a simple game by the set of minimal winning coalitions. A *dummy* player is a player that does not belong to any minimal winning coalition. Therefore, after eliminating dummy players from a game, the set of minimal winning coalitions does not change. As usual, we associate to a game a *dual* game. In the dual game, a coalition wins if its complement loses in the original game. As we will see later on, given a representation of a WMVG, a representation of its dual is easy to compute and therefore the enumeration avoids this kind of replica. Observe that as there are games that are self-dual, the number of WMVG without dummies is smaller than twice the number of WMVG without dummies and duals. In Table 1, we provided the known values for such subfamilies of WMVG.

In the quest for better algorithms to enumerate WMVG a lot of work has been devoted to find good representations. The first step is to restrict ourselves to integer representations, in which the weights of the players and the quota are integers. Freixas and Molinero [29] show that every WMVG admits an integer representation. Furthermore, they analyze which conditions can be added to restrict the considered representations. In particular, they introduce the *integer minimum* representation, in which the vector of weight is minimum in component-wise order. Ideally one would like to have a unique representation. As there are weighted majority voting games that do not admit a minimum representation [29], another notion of minimality, *minimum sum*, has been considered. In a minimum sum representation, the sum of the players' weights is minimum. Although WMVG do have minimum sum representations, there are games with more than one minimum sum representation [29,32,34]. To represent a game in a unique way, we follow the methodological approach used in the enumeration algorithm devised in [35] and consider what we call *canonical minimum* representations. This canonical minimum representation selects the lexicographically minimum sum representation of a WMVG.

Checking that a representation is minimum or minimum sum is a computationally hard problem as it involves the solution of integer linear programs [29]. Furthermore, few minimum or minimum sum representations of games with a large number of players are

known. However, it might be possible to get good representations from the distribution of weights and quotas in the target representations. Nevertheless, to the best of our knowledge nothing is known about such a distribution. One of the fundamental questions in the analysis of weighted voting games is to determine the relation among weight and power. This question has been addressed for particular classes of random weighted voting games obtained according to a fixed distribution of weights (see for example [36–40]). Having access to a good approximation to the real distribution of the player's weight might allow us to use the techniques on these papers to analyze the relation among power and weight on the complete set of weighted voting games.

Another natural approach to tackle the problem is to find procedures that allow the introduction of more players, while guaranteeing the optimality of the representation. Of course, due to the difficulty of the problem, it is also worth to analyze the optimality of representations, with many players, when only small numbers are allowed to be part of the representation.

With these three goals in mind, we started analyzing the list of canonical minimum representations of WMVG for up to eight players generated in Freixas and Molinero [29]. Using these data, we carried on a study of the distribution of the players' weights and the quotas in such representations. Although the estimated d istributions differ from the actual distributions, they become more similar as the number of players increase. Our results hint towards some promising probability distributions for larger numbers of players. As we will see, it can be inferred some tendency in the plots towards a Poisson or a $\chi^2$-Pearson distribution. We also observed that the range of weight values or quotas is contiguous up to 7 players but it becomes discontiguous for 8 players. In particular, weight 41 does not appear in any canonical minimum representation of games with 8 players, while 40 and 42 do. The results of this study are presented in Sections 3 and 4. For the second proposed line of research, we found several simple operations that allow us to construct minimum and, in some cases, minimum sum representations, of WMVG with many players. The corresponding construction and optimality proofs can be found in Section 5. Finally, we analyze games with a canonical minimum representation with small quotas. In particular, we show that for fixed $q$, the number of such games is upper bounded by a polynomial with degree $q$. Furthermore, we can show that for $q \leq 3$, all canonical minimum representations are minimum, independently of the number of players. Using this property, we show that the bound is tight for $q \leq 3$. Section 6 presents these results.

## 2. Definitions and Preliminaries

We use standard set theory notation. We follow definitions and notation for simple games from [41,42].

A *simple game (SG)* is a pair $(N, \mathcal{W})$ in which $N$ is a finite set of players and $\mathcal{W}$ is a monotone collection of subsets of $N$. We assume that $N = [n] = \{1, 2, \ldots, n\}$, that $\varnothing \notin \mathcal{W}$ and $N \in \mathcal{W}$. In terms of SG a subset $S \in \mathcal{W}$ is called a winning coalition and a subset $S \notin \mathcal{W}$ a losing coalition. We use $\mathcal{L}$ to denote the set of losing coalitions. Given a simple game $\Gamma = (N, \mathcal{W})$, a *minimal winning coalition* is a coalition in which the absence of any of the players present in the coalition turns the coalition losing. The set of minimal winning coalitions is denoted by $\mathcal{W}^m$. In the same way, a *maximal losing coalition*, denoted by $\mathcal{L}^M$, is a coalition such that by adding any new player the coalition becomes winning.

A simple game $\Gamma = (N, \mathcal{W})$ is a *weighted majority voting game (WMVG)* if there exists a $n + 1$-vector $[q; \mathbf{w}] = [q; w_1, w_2, \ldots, w_n]$, such that $S \in \mathcal{W}$ if and only if $\sum_{i \in S} w_i \geq q$. Due to monotonicity, we can assume that $w_i \geq 0$, for $i \in N$, and $q > 0$ because $\varnothing \notin \mathcal{W}$. The values in $\mathbf{w}$ are called the players' weights and the value $q$ the quota. Moreover, given $S \subseteq N$, $\mathbf{w}(S)$ denotes $\sum_{i \in S} w_i$. Thus, $q \leq \mathbf{w}(N)$ because $N \in \mathcal{W}$.

Observe that an assignment of players' weights and a quota define in a unique way the set of winning coalitions. When $\Gamma$ is a WMVG, we usually define $\Gamma$ by a representation $[q; \mathbf{w}]$. It is well known that every WMVG admits an *integer* representation, i.e., a $n + 1$-vector $[q; w_1, w_2, \ldots, w_n]$ in which all the values are non negative integers [41]. In the remaining

of the paper, we assume that all representations of WMVG are integer representations. Observe a WMVG admits more that one representation.

**Example 1.** *The representations* $[3; 1, 2, 1]$, $[6; 2, 4, 2]$ *and* $[10,000; 1, 9999, 1]$ *define the same simple game having* $\mathcal{W}^m = \{\{1, 2\}, \{2, 3\}\}$.

In fact, every WMVG admits infinitely many representations as $[q; \mathbf{w}]$ and $[cq; c\mathbf{w}]$, for any $c > 0$, represent the same game. In order to generate WMVG, we trim the number of possible representations of a game considering minimum representations.

**Definition 1.** *A representation* $[q; w_1, w_2, \ldots, w_n]$ *of a WMVG* $\Gamma$ *is said to be* minimum *if, for any representation* $[q'; w'_1, w'_2, \ldots, w'_n]$ *of* $\Gamma$, *we have that* $w_i \leq w'_i$, *for* $i \in N$.

Freixas and Molinero [29] have shown that not all WMVG admit a minimum representation. In particular, they listed the 154 WMVG with 8 players that have no minimum representation. However, if it exists, it is indeed unique. Another way of limiting the number of representations is by minimizing the sum of the players' weights.

**Definition 2.** *A representation* $[q; w_1, w_2, \ldots, w_n]$ *of a WMVG* $\Gamma$ *is a* minimum sum *representation if, for any representation* $[q'; w'_1, w'_2, \ldots, w'_n]$ *of* $\Gamma$, *we have* $\sum_{i=1}^{n} w_i \leq \sum_{i=1}^{n} w'_i$.

Although every WMVG admits a minimum sum representation this is not always unique. The 154 games with eight players mentioned before all have two minimum sum representations [29]. Observe that a minimum representation, if it exists, is also a minimum sum representation.

We say that player $i$ is a *dummy* player in $\Gamma = (N, \mathcal{W})$ if, for any $S \in \mathcal{W}$, then $S \setminus \{i\} \in \mathcal{W}$. Freixas and Molinero [29] proved the following useful results.

**Proposition 1.** *Let* $[q; \mathbf{w}]$ *be a minimum sum representation of a WMVG* $\Gamma$ *with n players.*

1. $q = \min_{S \in \mathcal{W}} \mathbf{w}(S)$ *and* $q = 1 + \max_{S \in \mathcal{L}} \mathbf{w}(S)$.
2. $w_i = 0$ *if and only if player* $i$ *is a dummy player.*
3. $gcd(q, \{w_i \mid w_i \neq 0\}) = 1$.

Furthermore, we have the following property.

**Proposition 2.** *Let* $[q; \mathbf{w}]$ *be a minimum sum representation of a WMVG* $\Gamma$. *For any non-dummy player* $i$, *there is at least one minimal winning coalition* $S$ *having* $i \in S$ *and* $\mathbf{w}(S) = q$.

**Proof.** Let $[q; w_1, \ldots, w_n]$ be a minimum sum representation of $\Gamma$. Assume that player $i$ has weight $w_i > 0$ and that no winning coalition $S$ with $w(S) = q$ contains $i$. Consider the game $\Gamma' = [q; w_1, w_2, \ldots, w_{i-1}, w_i - 1, w_{i+1}, \ldots, w_n]$. Observe that $\Gamma'$ has the same set of minimal winning coalitions as $\Gamma$, so $\Gamma$ admits a representation with smaller total weight and we get a contradiction. $\square$

Observe that, in minimum sum representations of games without dummy players, a coalitions $S$ with $\mathbf{w}(S) = q$ is a minimal winning coalition and, analogously, a coalition $S$ with $\mathbf{w}(S) = q - 1$ is a maximal losing coalition. The converses are not true. In fact, $\Gamma = [8; 4, 3, 3, 2, 2]$ verifies that $S = \{1, 2, 3\}$ is a minimal winning coalition such that $\mathbf{w}(S) = 10 > q$, and $T = \{2, 3\}$ is a maximal losing coalition such that $\mathbf{w}(T) = 6 < q - 1$. Moreover, all games without a minimum representation given by Freixas and Molinero [29] are also some counterexamples.

From the previous results, we can exclude representations of games with dummy players by considering only minimum sum representation $[q; \mathbf{w}]$ in which all the players' weights are positive, i.e., $\mathbf{w} > \mathbf{0}$. Besides, a dummy player never is part of a minimal winning coalition. So, after eliminating a dummy player from a WMVG the set of minimal

winning coalitions does not change. In this way, games with dummy players can be recovered from games without dummies with smaller number of players. In view of this fact, in the remaining of the paper, we consider only simple games without dummies.

Finally, observe that, by rearranging the players, we get isomorphic games. We are interested in defining a *unique* representation in the sense that two conceptually identical games, i.e., two isomorphic games, have the same representation. For doing so, we avoid rearrangements of the players by imposing an order on the players' weights.

**Definition 3.** $[q; w_1, w_2, \ldots, w_n]$ *is a* canonical representation *of* $\Gamma$ *if and only if* $w_i \geq w_j$ *whenever* $i < j$.

Canonical representations limit the number of possible representations but still do not provide a unique representation. For example, $[3; 2, 1, 1]$ and $[10{,}000; 999, 1, 1]$, are canonical representations of the SG with $\mathcal{W}^m = \{\{1, 2\}, \{1, 3\}\}$.

Note that all WMVG with 8 players and two minimum sum representations admit just one canonical representation [29]. For instance, $[25; 7, 6, 6, 4, 4, 4, 3, 2]$ and $[25; 7, 6, 6, 4, 4, 4, 2, 3]$ are minimum sum representations of the same game, but both representations lead to the same canonical representation $[25; 7, 6, 6, 4, 4, 4, 3, 2]$.

However, there are games with several minimum sum representations leading to different canonical representation. For instance, all examples with 10 and 11 players given by Freixas and Molinero [29]. As an example, consider the following representations $[68; 38, 31, 28, 23, 11, 8, 6, 5, 3, 1]$ and $[68; 37, 31, 28, 23, 11, 8, 7, 5, 3, 1]$, they define the same game, and both are minimum sum and canonical.

To perform our study, we keep just one minimum sum representation for each game (up to isomorphism) as follows.

**Definition 4.** $[q; \mathbf{w}]$ *is a* canonical minimum representation *of* $\Gamma$ *if* $[q; \mathbf{w}]$ *is canonical, minimum sum and, furthermore,* $\mathbf{w}$ *is lexicographically minimum among all players' weight vectors in canonical minimum sum representations of* $\Gamma$.

By selecting the lexicographically minimum, we guarantee that the representation is unique for each class of isomorphic games. For instance, the minimum sum and canonical representations $[68; 38, 31, 28, 23, 11, 8, 6, 5, 3, 1]$ and $[68; 37, 31, 28, 23, 11, 8, 7, 5, 3, 1]$ have the same canonical minimum representation, $[68; 37, 31, 28, 23, 11, 8, 7, 5, 3, 1]$.

Besides considering isomorphic games as equivalent, we use duality to drop even more the number of games to be considered. In this way, we also limit the number of possible game representations. Recall, that for a simple game $\Gamma = (N, \mathcal{W})$, its *dual game* is defined as $\Gamma^d = (N, \mathcal{W}^d)$ where $\mathcal{W}^d = \{S \mid N \setminus S \notin \mathcal{W}\}$. Furthermore, the dual of $\Gamma^d$ is $\Gamma$. We call a game *self-dual* if $\Gamma = \Gamma^d$. In the case of a WMVG, we can obtain a minimum sum representation of the dual from a minimum sum representation.

**Proposition 3.** *Let* $[q; \mathbf{w}]$ *be a minimum sum representation of* $\Gamma$*, then* $[\mathbf{w}(N) - q + 1; \mathbf{w}]$ *is a minimum sum representation of* $\Gamma^d$ .

**Proof.** Let us prove first that the weighted voting game $\Gamma' = [\mathbf{w}(N) - q + 1; \mathbf{w}]$ is indeed $\Gamma^d$. Consider a set $S \subseteq N$, $S$ wins in $\Gamma'$ if and only if $\mathbf{w}(S) \geq \mathbf{w}(N) - q + 1$. However, then, $\mathbf{w}(N \setminus S) = \mathbf{w}(N) - \mathbf{w}(S) < q$. Therefore, $S$ wins in $\Gamma'$ if and only if $N \setminus S$ loses in $\Gamma$. As the weights of the players are the same in both representations, and $\Gamma$ is the dual of $\Gamma^d$, the transformed representation is also a minimum sum representation. $\square$

The previous result establishes that the representations of dual games have the same weights. Therefore, their canonical minimum representations differ only in the value of the quota. In order to keep a unique representation, up to isomorphism and duality, we restrict also the value of the quota.

**Definition 5.** $[q; \mathbf{w}]$ *is a* strict canonical minimum representation *of* $\Gamma$ *if* $[q; \mathbf{w}]$ *is a canonical minimum representation and* $q \geq \frac{\mathbf{w}(N)+1}{2}$.

Observe that by using only strict canonical minimum representations, we count as one any pair of mutually dual games. Thus, the number of canonical minimum representations is smaller than twice the number of strict canonical minimum representations due to the representations of self-dual games. In fact, the number of all WMVG is equal to the number of self-dual WMVG plus twice the number of non self-dual strict canonical minimum representations.

## 3. Weights in WMVG up to Eight Players

Our study is grounded on the data provided by Freixas and Molinero [35]. The data set contains all the canonical minimum representations of a WMVG (without dummies) up to eight players. Our first objective is to analyze the weights appearing in the canonical minimum representations. We started from four data sets. Each data set is formed by the canonical minimum representations of WMVG without dummies ($[q; \mathbf{w}]$ having $\mathbf{w} > \mathbf{0}$), for $n = 5, \ldots, 8$. For smaller values of $n$, we just did the computations by hand. All the data obtained in our study can be found in Appendix A.

Before presenting the results let us introduce some notation. We use $w_n^{\max}$ to denote the maximum weight appearing in a canonical minimum representation of WMVG with $n$ players. As we are considering games without dummy players, and $[1; 1, \ldots, 1]$ is a minimum representation, the corresponding minimum weight is 1, for any $n$. We denote by $w_n^{\text{u-min}}$, the minimum non-repeated (*unique*) weight, i.e., the smallest weight that appears in some canonical minimum representation but that never appears more than once in a canonical minimum representation. Finally, we say that a weight $x \in [w_n^{\max}]$ is a *skip* if $x$ does not appear in any canonical minimum representation of games with $n$ players.

In our first experiment, we perform an analysis of the weights appearing in the canonical minimum representations. For doing so, we create new data sets, for $n = 5, \ldots, 8$, containing the concatenation of all the weight vectors appearing in the initial data sets, for the corresponding number of players. We refer to those data sets as *weights* in canonical minimum representations.

We start our study analyzing, for each weight and value of $n$, some basic features. The results are summarized in Table 2. The first interesting property we observed is that, for $n < 8$, there are no skips among the weights. However, this property does not hold for 8 players. In particular, there is no canonical minimum representation with eight players holding weight 41, although there are such representations with weight 42, and with weights $1, \ldots, 40$. Furthermore, Freixas and Molinero [29] provided the minimum sum representations of games with eight players and without minimum representation. None of these minimum sum representations includes value 41. Therefore, the canonical minimum sum representations, for eight players, have a skip. Note that, as we will see later on, there are canonical minimum representations with more than eight players holding weight 41.

**Table 2.** Features of the weights in canonical minimum representations.

| # of Players | $w_n^{\text{u-min}}$ | $w_n^{\max}$ | Mode | Skips |
|---|---|---|---|---|
| 1 | 1 | 1 | 1 | None |
| 2 | None | 1 | 1 | None |
| 3 | 2 | 2 | 1 | None |
| 4 | 3 | 3 | 1 | None |
| 5 | 4 | 5 | 1 | None |
| 6 | 6 | 9 | 2 | None |
| 7 | 10 | 18 | 2 | None |
| 8 | 19 | 42 | 3 | 41 |

We can also observe that $w_n^{\mathrm{max}}$ seems to grow at least exponentially, while the mode of the weights seems to grow sub-linearly. For $n = 2$, the canonical minimum representations of the two WMVGs with two players are $[1; 1, 1]$ and $[2; 1, 1]$. This is the unique case in which $w_n^{\mathrm{u\text{-}min}}$ is undefined. Another interesting property is that, for $3 \leq n \leq 8$, $w_n^{\mathrm{u\text{-}min}} = w_{n-1}^{\mathrm{max}} + 1$. These values for $w_n^{\mathrm{max}}$ coincide with those obtained by Kurz [32].

In Figure 1, we plot the attained frequency distributions for $n \geq 5$ only, as the results, for up to 4 players, do not provide any information because the number different weights is small.

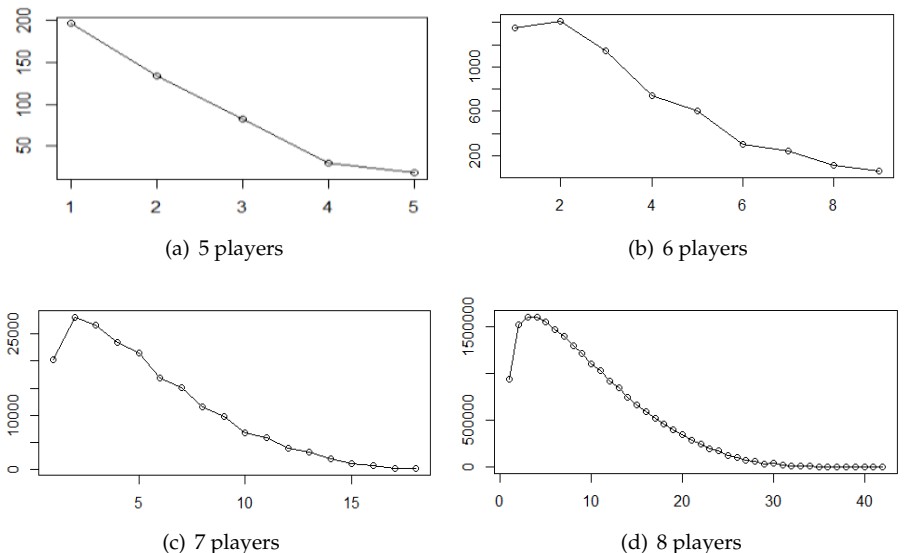

(a) 5 players                 (b) 6 players

(c) 7 players                 (d) 8 players

**Figure 1.** Frequency distribution of weights in canonical minimum representations.

There are some noteworthy things to remark. Visually checking the plots, it can be seen that, as the number of players increases, so does the smoothness of the distribution peaking at Figure 1d. Furthermore, although unexpected, close weights have a close number of occurrences.

Due to the similarities in the plots and therefore in their distributions, we suspect that a pattern might be present. In order to further study the mentioned similarities, we take a standard probabilistic approach, Kernel density estimation [43]. The method provides a tool to smooth the data representation based on a finite data sample. In Figure 2, we present an estimation of the density function. We obtained this approximation using a built-in function in the R language for Kernel density estimation. We run the procedure with the default values and selecting a Gaussian Kernel. The method basically puts a Gaussian over each data point and sums up the densities (with proper normalisation). The values on the *x*-axis correspond to this normalized sum, observe that due to the tails of the Gaussians the left and the right limits are increased.

The tendency here is clear, at the beginning a mixture of two Gaussians is present, but as the number of players increases one dominates over the other slowly turning the mixture to a normal distribution density. Furthermore, the weight that appeared the most is slowly increasing as the number of players increases. It seems that weights will tend towards a normal distribution.

Our second analysis focuses on the weights in strict canonical minimum representations. For doing so, we create new data sets, for $n = 5, \ldots, 8$, containing the concatenation of all the weight vectors in canonical representations $[q; \mathbf{w}]$ having $q \geq \frac{\mathbf{w}(n)+1}{2}$. We refer to those data sets as *weights* in strict canonical minimum representations. Here, the main difference is that, if a game is not self-dual, their weights are counted once while they were counted twice before. Our aim is to determine whether self-dual games intervene strongly in the distribution or not.

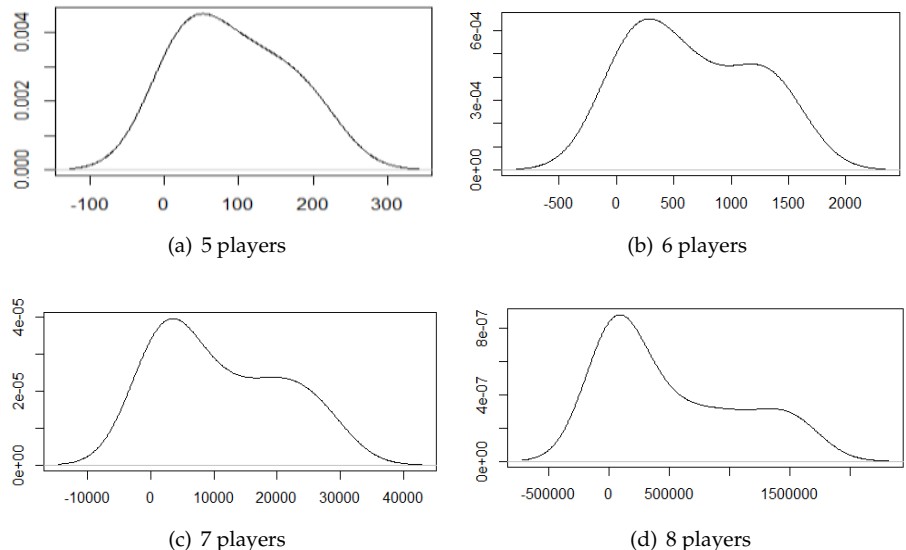

**Figure 2.** An estimation of the density function of weights in canonical minimum representations.

The plots of the frequency distribution of weights in strict canonical minimum representations are given in Figure 3. We can see that the frequency distributions are practically the same as in Figure 1. This seems to reflect the sparsity of self-dual games, especially as the number of players increases.

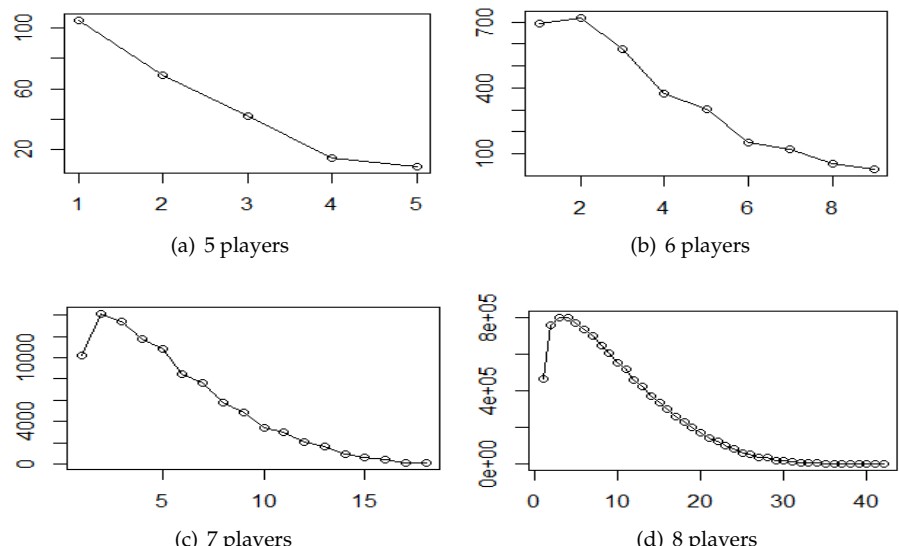

**Figure 3.** Frequency distribution of weights in strict canonical minimum representations.

Comparing the plots in Figures 1 and 3, it can be observed that the frequency of each weight seems to be around one half of the ones in the the canonical minimum representations. This is according to the the sparsity of the self-dual WMVG. Hence, we get about the same distributions, densities and features, but with a reduction in the frequencies by a factor of around $\frac{1}{2}$. Our results indicate that, as the number of players increases, the relevance of self-dual games decreases.

Our last step is to study the distribution of the frequencies of weights in canonical minimum representations disregarding multiplicities. For doing so, we create data sets, for $n = 5, \ldots, 8$, including, from each canonical minimum representation $[q; \mathbf{w}]$, the set of weights appearing in $\mathbf{w}$. Now, if a canonical minimum representation repeats weights, then

each weight is included only once in the data set. The corresponding plots are depicted in Figure 4.

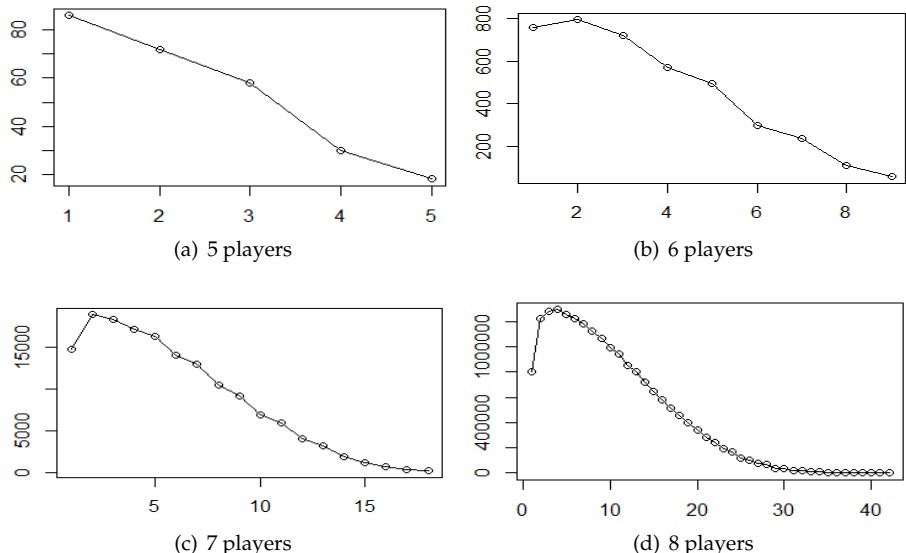

**Figure 4.** Frequency distribution of weights without multiplicity in canonical minimum representations.

Again, we can observe that the frequency distributions are like the previous ones but approximately scaled by a constant. This seems to indicate the average frequency is constant. Something to remark is that the mode remains the same for $n < 8$, and increases to 4 when $n = 8$. This, in some sense, shows that a weight is not particularly frequent because it appears multiple times. Furthermore, we can see that the right tails of Figure 1a–d and those in Figure 4a–d look much the same. In other words, if the weight was large enough, then the weight appears at most once.

## 4. Quotas in WMVG up to 8 Players

Our second study concerns the quotas for games up to eight players. In this case, we first study the distribution of the frequencies of quotas in canonical minimum representations. For doing so, we create data sets, for $n = 5, \ldots, 8$, including, from each canonical minimum representation $[q; \mathbf{w}]$ of games with $n$ players, the value $q$. Similar as for the weights analysis, $q_n^{\max}$ denotes the maximum quota appearing in a canonical minimum representation, and a skip is a quota value that does not appear in any representation.

The most relevant information on these data sets is summarized in Table 3.

**Table 3.** Features of the distribution of the quotas in canonical minimum representations.

| # of Players | $q_n^{\max}$ | Mode | Skips |
|---|---|---|---|
| 1 | 1 | 1 | None |
| 2 | 2 | 1, 2 | None |
| 3 | 3 | 2, 3 | None |
| 4 | 5 | 3, 4 | None |
| 5 | 9 | 5 | None |
| 6 | 18 | 11 | None |
| 7 | 40 | 19 | None |
| 8 | 105 | 37 | 95, 97, 99, 100, 101, 103, 104 |

One can observe that the growth of the maximum quota value seems to be at least exponential, and that it is much faster than the growth of the maximum weight (see Table 2). As for the weights, no skips in the quota values appear for less than 8 players. However,

seven quotas are not present, for $n = 8$. Furthermore the skipped quotas are not contiguous. The mode of the quota's data sets seem to grow at least exponentially, too. An interesting trait here, is that the mode of the quotas, for 5 or more players, are all prime numbers. It will be of interest to know whether this property carries over to higher number of players.

Similar to the study of the weights, we study the distribution and density functions for the frequency of the quota values. The results, for WMVG with 5, 6, 7 and 8 players, are depicted in Figures 5 and 6. We can see that the distribution plots depict what seems to be a "smooth" function. The distributions are symmetrical, with equal tails, and, in general, look like a normal distribution.

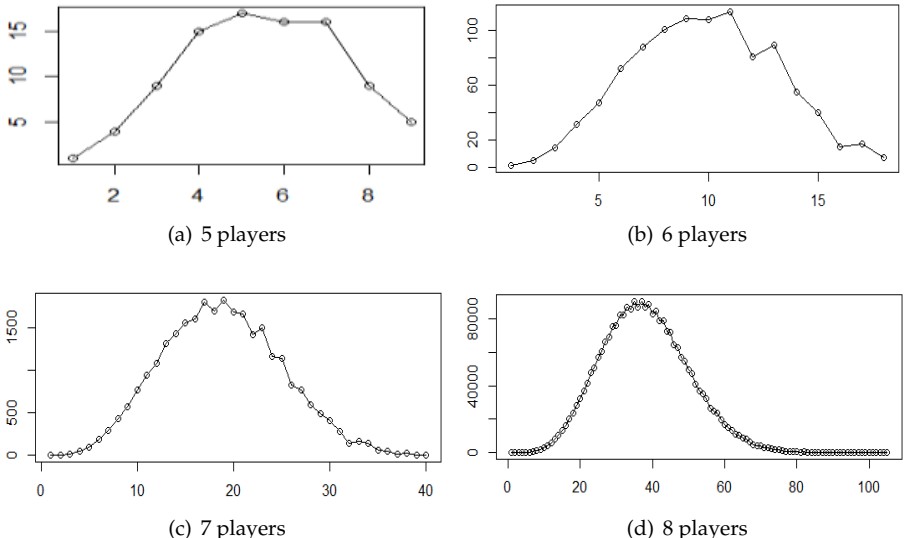

**Figure 5.** Distribution of quota occurrences in canonical minimum representations.

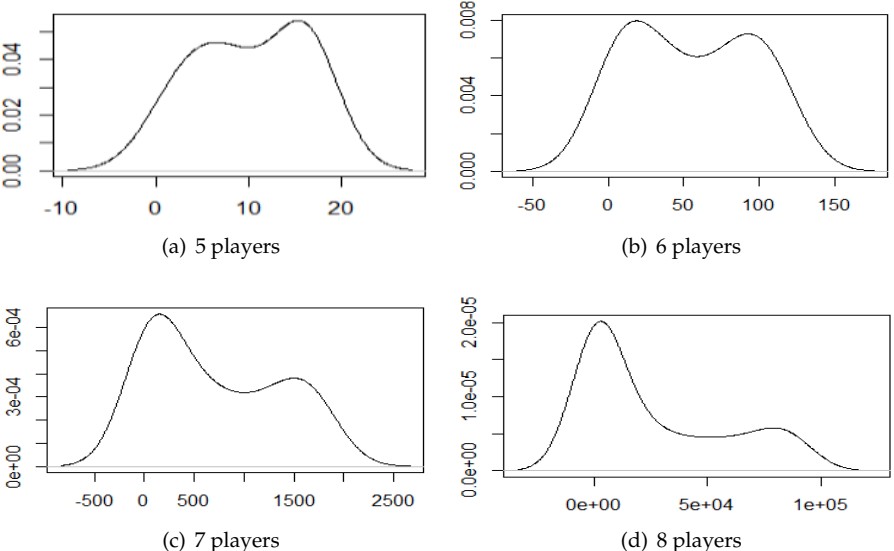

**Figure 6.** An estimation of the density function of quotas in canonical minimum representations.

The densities' plots, like in the case of weights, look like a mixture of two Gaussians. One of the Gaussians slowly disappears as the number of players increases, transforming into what looks like a regular normal distribution.

## 5. Generating Minimum and Minimum Sum Representations

In this section, we show how some canonical minimum and minimum representations of games with any number of players can be obtained. For doing so we analyze several ways to introduce players into a game in such a way that the extended representation can be proved to be minimum or minimum sum.

Given a general WMVG, the generation of a minimum, minimum sum, or canonical minimum representation is a computationally hard task. Recall, that deciding if a player is a dummy in a WMVG is a NP-complete problem [42], while this problem can be solved easily given a minimum sum representation.

To obtain new minimum/minimum sum representations of games with many players we consider the following operation. Let $\mathbf{w}$ be a weight assignment for $n$ players. For a non-negative $x$, $x \oplus \mathbf{w}$ is the weight assignment obtained from $\mathbf{w}$ by adding component $n + 1$ holding weight $x$. When dealing with canonical representations, we will assume that the components of $x \oplus \mathbf{w}$ are rearranged in such a way that it is canonical for the $n + 1$ players.

The following result gives equivalent representations under some conditions.

**Lemma 1.** *Let $[q; \mathbf{w}]$ and $[q + t; t \oplus \mathbf{w}]$ be representations of $\Gamma = ([n], \mathcal{W})$ and $\Gamma' = ([n + 1], \mathcal{W}')$, respectively. If, for some $k \neq t$, $[q'; k \oplus \mathbf{w}']$ is another representation of $\Gamma'$, then $[q' - k; \mathbf{w}']$ is a representation of $\Gamma$.*

**Proof.** The coalition $S$ is winning in $\Gamma$ is and only if coalition $S \cup \{n + 1\}$ is winning in $\Gamma'$, and hence $S \in \mathcal{W} \iff \mathbf{w}'(S \cup \{n + 1\}) \geq q' \iff \left(\sum_{i=1}^{n} w_i'\right) + w_{n+1} \geq q' \iff \sum_{i=1}^{n} w_i' \geq q' - w_{n+1}$. □

We start presenting some procedures leading to minimum sum representations.

**Proposition 4.** *Let $[q; \mathbf{w}]$ be a minimum sum representation of a WMVG with $n$ players, then the representations $[q; 0 \oplus \mathbf{w}], [q; 1 \oplus \mathbf{w}]$ and $[q + 1; 1 \oplus \mathbf{w}]$ are minimum sum representations.*

**Proof.** In the procedure to get a game with representation $[q; 0 \oplus \mathbf{w}]$, we add a dummy player. By adding a dummy player, the minimal winning coalitions do not change. So, if $[q; 0 \oplus \mathbf{w}]$ is not minimum sum, there is a way to represent with less weight sum the created game. As a dummy player in a minimum sum representation gets weight 0 this representation has the form $[q'; 0 \oplus \mathbf{w}']$. Then $[q'; \mathbf{w}']$ will be a representation for the original $n$ players game with less. Therefore, $[q; 0 \oplus \mathbf{w}]$ is a minimum sum representation.

In $[q; 1 \oplus \mathbf{w}]$, we are adding player $n + 1$ with weight 1. As the quota is not changed, the minimal winning coalitions in $[q; \mathbf{w}]$ are also minimal winning coalitions in $[q; 1 \oplus \mathbf{w}]$. At the same time, all the losing coalitions with $\mathbf{w}(A) = q - 1$ become minimal winning coalitions in $[q; 1 \oplus \mathbf{w}]$ with the help of the new player. From Proposition 1, we know that there is at least one such coalition, so the set of minimal winning coalitions changes.

Suppose that $[q; 1 \oplus \mathbf{w}]$ is not minimum sum, meaning that there exists another representation $[q'; w_1', \ldots, w_n', w_{n+1}']$ with $\sum_{i=1}^{n+1} w_i' < \sum_{i=1}^{n+1} w_i$ and $w_{n+1}' \geq 1$ representing the same game. Observe that $w_{n+1}'$ cannot be 0 as this leads to a game with a different set of minimal winning coalitions. As $w_{n+1}' \geq 1$, the reduction in weight has to be done in the first components, but this implies that we can represent $[q; \mathbf{w}]$ with less total weight, contradicting the fact that it was a minimum sum representation.

Let us look at the winning coalitions in $[q + 1; 1 \oplus \mathbf{w}]$. Those are $S \subseteq [n]$ with $\mathbf{w}(S) > q$ or $S \cup \{n + 1\} \subseteq [n + 1]$ with $\mathbf{w}(S) = q$. Note that since there is a coalition with weight $q$ in $[q; \mathbf{w}]$, player $n + 1$ is not dummy. Hence its weight is necessarily positive in any minimum sum representation.

Suppose that $[q + 1; 1 \oplus \mathbf{w}]$ is not minimum sum. In this case, there is a representation $[q'; w_1', w_2', \ldots, w_n', w_{n+1}']$ of the game with $\sum_{i=1}^{n+1} w_i' < \sum_{i=1}^{n} w_i + 1$ and $w_{n+1}' \geq 1$. Therefore,

we have $\sum_{i=1}^{n} w_i' < \sum_{i=1}^{n} w_i$. Now, $[q' - w_{n+1}'; w_1', w_2', \ldots, w_n', 0]$ has the same minimal winning coalitions than $[q; \mathbf{w}]$ and less total sum, contradicting our initial assumption. $\square$

Our next result shows how the set of procedures can be extended when the objective is to get minimum representations from a given minimum representation.

**Proposition 5.** *Let $[q; \mathbf{w}]$ be a minimum representation of a WMVG with n players, then the representations $[q; 0 \oplus \mathbf{w}]$, $[q; 1 \oplus \mathbf{w}]$, $[q + 1; 1 \oplus \mathbf{w}]$, $[q; q \oplus \mathbf{w}]$ and, for $i \in [n]$, $[q; w_i \oplus \mathbf{w}]$ and $[q + w_i; w_i \oplus \mathbf{w}]$ are minimum representations.*

**Proof.** Recall that if a representation is minimum it is also the unique minimum sum representation of the corresponding game.

From Proposition 4, we know that $[q; 0 \oplus \mathbf{w}]$, $[q; 1 \oplus \mathbf{w}]$ and $[q + 1; 1 \oplus \mathbf{w}]$ are minimum sum representations. Furthermore, any other minimum sum representation of these games has to keep the weight of player $n + 1$ as 0 in the first case and positive in the other two cases. Therefore, removing the player from the respective game will lead to a different minimum sum representation of $[q; \mathbf{w}]$, contradicting our hypothesis.

Assume that $[q; q \oplus \mathbf{w}]$ is not minimum. In such a case, there is another representation $[q'; w_{n+1}' \oplus \mathbf{w}']$ such that at least one weight in $w_{n+1}' \oplus \mathbf{w}'$ is smaller than in $q \oplus \mathbf{w}$. Furthermore, we can assume that $[q'; w_{n+1}' \oplus \mathbf{w}']$ is a minimum sum representation. In the game $\Gamma'$ represented by $[q; q \oplus \mathbf{w}]$, the coalition $\{n + 1\}$ is winning. Therefore, in any minimum sum representation of $\Gamma'$, the weight of player $n + 1$ must coincide with the quota. So, $w_{n+1}' = q'$. Moreover, $[q'; \mathbf{w}']$ and $[q; \mathbf{w}]$ represent the same game $\Gamma$. Now we consider two cases.

**Case $q' \geq q$:** As $[q'; q' \oplus \mathbf{w}']$ is a minimum sum representation of $\Gamma'$, $q' + \mathbf{w}'([n]) \leq q + \mathbf{w}([n])$. Then $\mathbf{w}'([n]) \leq \mathbf{w}([n])$, so $[q; \mathbf{w}]$ is not minimum.

**Case $q' < q$:** As $[q; \mathbf{w}]$ is a minimum representation, $\mathbf{w}$ is component wise smaller than $\mathbf{w}'$. So, for $S \subseteq [n]$, $\mathbf{w}'(S) \geq \mathbf{w}(S)$. As $[q; \mathbf{w}]$ represents $\Gamma$, for all the winning coalitions $S \subseteq [n]$ of $\Gamma$, we have $\mathbf{w}(S) \geq q > q'$. Furthermore, as $[q'; \mathbf{w}']$ represents $\Gamma$, for all losing coalitions $S \subseteq [n]$ of $\Gamma$, $\mathbf{w}'(S) < q'$ and thus, $\mathbf{w}(S) \leq \mathbf{w}'(S) < q'$. Therefore, $[q'; q' \oplus \mathbf{w}]$ and $[q'; q' \oplus \mathbf{w}']$ represent the same game. However, $q' + \mathbf{w}'([n]) \geq q' + \mathbf{w}([n])$, contradicting the fact that $[q'; q' \oplus \mathbf{w}']$ was a minimum sum representation.

For the procedures with weights, assume that $w_i \oplus \mathbf{w}$ does not provide the minimum representation weights. In such a case, there is another representation $[q'; w_{n+1}' \oplus \mathbf{w}']$ with players' weight $\mathbf{w}'$ such that at least one weight in $\mathbf{w}'$ is smaller than in the first one. Observe that $[q'; w_{n+1}' \oplus \mathbf{w}']$ is not required to be a minimum representation. If at least one player with weight $w_i$ keeps or does not reduce its weight, we can assume w.l.o.g. that this player is player $n + 1$. Observe that, in $w_i \oplus \mathbf{w}$ there are at least two players with weight $w_i$. Therefore, our assumption guarantees that there exists $j \in [n]$ such that $w_j' < w_j$.

Suppose that $[q; w_i \oplus \mathbf{w}]$ is not minimum. Consider the representation $[q'; w_{n+1}' \oplus \mathbf{w}']$ for the $\mathbf{w}'$ described before. Observe that, by construction, the minimal winning coalitions of $[q; \mathbf{w}]$ and $[q'; \mathbf{w}']$ coincide, therefore $[q; \mathbf{w}]$ and $[q'; \mathbf{w}']$ represent the same game. However, there is $j \in [n]$ such that $w_j' < w_j$. Therefore, $[q; \mathbf{w}]$ cannot be minimum, contradicting our hypothesis.

Suppose that $[q + w_i; w_i \oplus \mathbf{w}]$ is not minimum. Consider the representation $[q'; w_{n+1}' \oplus \mathbf{w}']$ described before. By Lemma 1, $[q; \mathbf{w}]$ and $[q' - w_{n+1}'; \mathbf{w}']$ represent the same game. However, as there is $j \in [n]$ such that $w_j' < w_j$, we reach a contradiction. $\square$

We want to point out that the converses of the former implications are in general false. Consider for example the representation $[7; 5, 2, 2, 1, 1]$. This is the unique minimum representation of the game with minimal winning coalitions $\{\{1, 2\}, \{1, 3\}, \{1, 4, 5\}\}$. Deleting any of the repeated weights (1 or 2) yields a game with 4 players but with maximum weight 5. From Table 2, we know that the maximum weight present in canonical minimum

representations for games up to 4 players is 3. Therefore, independently of the value of $q$, the representation of the game with 4 players (and the same weights) is not minimum.

The results in Proposition 4 allow us to generate minimum representations with any number of players applying iteratively the procedures. For example, from the minimum representation $[41; 12, 11, 10, 8, 4, 4, 2, 1]$ with 8 players, applying each of the procedures $i > 0$ times, we generate the following minimum representations of games with $8 + i$ players,

$$[41; 12, 11, 10, 8, 4, 4, 2, 1, 0{:}i],$$
$$[41; 12, 11, 10, 8, 4, 4, 2, 1, 1{:}i],$$
$$[41 + i; 12, 11, 10, 8, 4, 4, 2, 1, 1{:}i],$$
$$[41; 41{:}i, 12, 11, 10, 8, 4, 4, 2, 1],$$
$$[41; 12, 11, 10, 8, 8{:}i, 4, 4, 2, 1],$$
$$[41 + 8i; 12, 11, 10, 8, 8{:}i, 4, 4, 2, 1].$$

In the above representations we use $w{:}i$ to indicate that there are $i$ players with assigned weight $w$. Of course, we can also mix different procedures getting, for example, the minimum representation with $9 + i + j + k$ players

$$[41 + 41i + 8j + k; 41, 41{:}i, 12, 11, 10, 8, 8{:}j, 4, 4, 2, 1, 1{:}k].$$

A basic question is whether a skip value (weight or quota) can appear in games with higher numbers of players. From our data, 41 is a weight skip but it is not a quota skip. Using the above procedures, we can prove that any natural number appears as weight or as quota in a minimum representation for a big enough number of players.

**Corollary 1.** *For $q \geq 1$, there exists $n_q > 0$ and a minimum representation of a WMVG with $n_q$ players having quota $q$. For $w \geq 0$, there exists $n_w > 0$ and a minimum representation of a WMVG with $n_w$ players including weight $w$.*

**Proof.** Note that all canonical minimum representations with less than 8 players are minimum representations. Furthermore, weights and quotas of the games of 7 players are contiguous. In particular, there are minimum representations of games with 7 players and quota $q$, for $1 \leq q \leq 40$. If $q > 40$, we consider a minimum representation $[40; \mathbf{w}]$ of a game with 7 players. According to Proposition 4, $[40 + i; \mathbf{w} \oplus 1 : i]$ is a minimum representation. Taking $i = q - 40$, we get a minimum representation with quota $q$.

Therefore, using the fact that if $[q; \mathbf{w}]$ is a minimum representation then $[q; q \oplus \mathbf{w}]$ is also a minimum representation, we can extend the result to weights.  $\square$

The previous result shows that the weight and quotas skipped by games with 8 players appear in minimum representations of games with more than 8 players. In particular, the weight 41 appears in minimum representations of games with 9 players. However, for the skipped quotas, the number of players is at least 55. We can improve those bounds on the number of players needed for the quota skips in games with 8 players.

**Corollary 2.** *There is a minimum representation of a WMVG with 9 players in which one player has weight 41. There are minimum representations of WMVG with 12 players and quotas 95, 97, 100, 101, 103 and 104, and with 13 players and quotas 99.*

**Proof.** As we have shown before $[41; 41, 12, 11, 10, 8, 4{:}2, 2, 1]$ is a minimum representation of a game with 9 players and one player has weight 41.

From this representation, by adding players with repeated weights and increasing in the same amount the quota, we can generate the following minimum representations, with 12 players

$$[95; 41{:}2, 12{:}2, 11, 10, 8, 4{:}2, 2, 1{:}2],$$
$$[97; 41{:}2, 12, 11{:}2, 10, 8, 4{:}3, 2, 1],$$
$$[100; 41{:}2, 12, 11, 10{:}2, 8{:}2, 4{:}2, 2, 1],$$
$$[101; 41{:}2, 12, 11{:}2, 10, 8{:}2, 4{:}2, 2, 1],$$
$$[103; 41{:}2, 12, 11{:}2, 10{:}2, 8, 4{:}2, 2, 1],$$
$$[104; 41{:}2, 12, 11{:}3, 10, 8, 4{:}2, 2, 1],$$

and with 13 players:

$$[99; 41{:}2, 12, 11, 10, 8{:}3, 4{:}2, 2, 1{:}2].$$

□

There are more results that can be derived from Proposition 4. Recall that $w_n^{\max}$, $q_n^{\max}$ and $w_n^{\text{u-min}}$ are, respectively, the maximum weight, maximum quota, and minimum non-repeated weight in the canonical minimum representations of WMVGs with $n$ players. Consider now the corresponding values taken only over the minimum representations. We use $\overline{w}_n^{\max}$, $\overline{q}_n^{\max}$ and, respectively, $\overline{w}_n^{\text{u-min}}$ to represent these values. In Proposition 5 we have shown that, when $[q; \mathbf{w}]$ is a minimum representation, $[q; q \oplus \mathbf{w}]$ is also a minimum representation. Therefore, we have the following result.

**Corollary 3.** *For any $n > 1$, $\overline{q}_{n-1}^{\max} \leq \overline{w}_n^{\max}$.*

As we have mentioned before, all canonical minimum representations for $n \leq 7$ are minimum. For $n = 8$, we have games without minimum representations. For $n = 8$, after checking for minimality, we found that $\overline{w}_8^{\max} = w_8^{\max}$, $\overline{q}_8^{\max} = q_8^{\max}$ and $\overline{w}_8^{\text{u-min}} = w_8^{\text{u-min}}$. Looking at the values reported in Tables 2 and 3, the above inequality is tight for 1 to 7 players, and quite accurate for 8 and 9 players, taking into account that, according to [32], $w_9^{\max} = 110$.

In Proposition 5, we have shown that, if $[q; \mathbf{w}]$ is a minimum representation, then $[q; w_i \oplus \mathbf{w}]$ is also minimum. Therefore, the maximum weight appearing in a minimum representation with $n$ players, appears more than once in minimum representations with $n + 1$ players. Then, the following result holds.

**Corollary 4.** *For any $n > 2$, if $\overline{w}_n^{\boldsymbol{u}\text{-}\boldsymbol{min}}$ exists, then $\overline{w}_n^{\boldsymbol{u}\text{-}\boldsymbol{min}} > \overline{w}_{n-1}^{\max}$.*

From these results in Table 2, the inequality $\overline{w}_n^{\text{u-min}} > \overline{w}_{n-1}^{\max}$ is tight from 3 to 8 players as the corresponding values differ in one unit.

## 6. Small Quotas in Canonical Minimum Representations of WMVG

In this section, we analyze canonical minimum representations in which $q \leq 3$. Our first results are closed formulas for the number of WMVG without dummies having a canonical minimum representation with quota at most 3. In order to get the results we need to analyze some properties of such representations.

As a consequence of Proposition 1, we know that in a minimum sum representation $[q; \mathbf{w}]$, the weight of any player is at most $q$. Therefore, from the range of possible values, we have that the number of minimum sum representations of WMVG without dummies, with $n$ players, and quota $q$ is at most $q^n$. When dummies are allowed, the upper bound is $(q + 1)^n$. These bounds are far from optimal, the naive counting includes many combinations that are not canonical or minimum sum. For example, $[q; q, q \ldots, q]$ is not minimum sum as the game can be represented by $[1; 1, 1, \ldots, 1]$. Furthermore, we are counting as different isomorphic representations. To improve the upper bound we take a different approach.

Recall that a *weak composition* of a non-negative integer $n$ into $k$ parts is a $k$-tuple of non-negative integers that sum to $n$. For example, $(1,0,0,3)$ is a weak composition of four into four parts. Observe that, as the definition is in terms of tuples, by reordering the components we get a different composition. Now, take into account that the number of weak compositions is

$$
\begin{aligned}
\binom{n+k-1}{k-1} &= \frac{(n+k-1)!}{n!(k-1)!} = \frac{(n+k-1)(n+k-2)\cdots(n+1)}{(k-1)!} \\
&= \frac{1}{(k-1)!}n^{k-1}\underbrace{\left(1+\frac{k-1}{n}\right)}_{<2}\underbrace{\left(1+\frac{k-2}{n}\right)}_{<2}\cdots\underbrace{\left(1+\frac{1}{n}\right)}_{<2} \\
&\leq \frac{1}{(k-1)!}(2n)^{k-1}.
\end{aligned}
$$

We use standard O-notation for asymptotic upper bounds following Cormen et al. [44]. Thus, for a non-constant $k$, we have $\binom{n+k-1}{k-1} = O\left(\frac{(2n)^{k-1}}{(k-1)!}\right)$.

**Proposition 6.** *The number of canonical minimum representations of WMVG without dummies, with $n$ players and quota $q$, is $O\left(\frac{(2n)^{q-1}}{(q-1)!}\right)$. When dummies are allowed this number becomes $O\left(\frac{(2n)^q}{q!}\right)$.*

**Proof.** For games without dummies, we know that the possible values for the players' weights are between 1 and $q$. Consider the set of $q$-tuples $(A_1, A_2, \ldots, A_q)$ with $A_i \in \{0, \ldots, n\}$ and whose sum adds up to $n$. Such a tuple defines in a unique way a canonical representation with quota $q$ and having $A_i$ players with weight $i$. Hence, the number of such compositions of $n$ is an upper bound on the number of canonical minimum representations, and we get the upper bound.

For games without dummy players, the analysis is the same, taking into account that the value 0 can be present. This leads to compositions of $n$ into $q+1$ parts. $\square$

The previous result provides an upper bound on the number of canonical minimum representations with a given quota. However, we have no assurance that this bound is indeed tight. We will show that the bound is tight for $q \leq 3$. Before doing so we need an auxiliary result.

**Lemma 2.** *Let $[q; \mathbf{w}]$ be a canonical minimum sum representation of $\Gamma$. For $q \leq 3$, $[q; \mathbf{w}]$ is a minimum representation and hence we have uniqueness in the representation.*

**Proof.** We divide the proof into cases depending on the value of $q$.

**Case q = 1:** According to Proposition 1, any canonical minimum sum representation with quota 1 has the form $[1; 1, \ldots, 1, 0, \ldots, 0]$. As $\mathbf{w}(N)$ is the number of 1 s in $\mathbf{w}$, any other minimum sum representation of $\Gamma$ must have the same number of 1 s. Therefore, the representation is unique.

**Case q = 2:** In this case $\mathbf{w}$ must have the form $(2, \ldots, 2, 1, \ldots, 1, 0, \ldots 0)$. If another minimum sum representation exists it must have the same number of 0s. As the sum must be preserved, the only possibility is to increase some 1 weights to 2 and to balance these changes by decreasing the same number of 2 s and 1 s. However this transformation leads to the same canonical representation.

**Case q = 3:** Now $[q; \mathbf{w}]$ has the form $[3; 3, \ldots, 3, 2, \ldots, 2, 1, \ldots, 1, 0, \ldots, 0]$. Exactly as before, an other representation must have the same number of 0s. A symmetric argument shows that the total number of 3 s must be preserved. This is because $w_i = 3$ if and only if $\{i\}$ is a minimal winning coalition. Therefore the only possibility is that another representation is obtained by increasing by 1 some 1 s and decreasing by 1 some 2 s. As the sum must be preserved, as in the previous case, the corresponding canonical representations are identical.

$\square$

Now we know that if a game has a minimum sum representation with quota 1, 2 or 3 then this representation is minimum. Using this characterization, we are able to count exactly, for any $n$, the number of games with $n$ players having a canonical minimum representation with quotas up to 3.

Let $M(n, q)$ be number of WMVG without dummies with $n$ players having a canonical minimum representation with quota $q$ and let $D(n, q)$ be number of WMVG with $n$ players having a canonical minimum representation with quota $q$.

**Proposition 7.** *For any $n > 1$, $M(n, 1) = 1$, $M(n, 2) = n - 1$, $M(n, 3) = \frac{(n-2)(n+1)}{2}$.*

**Proof.** Let us analyze first the case in which dummies are not allowed. In this case, we know that the players must have positive weight, and that no player can have greater weight than the quota. When $q = 1$, the unique possible representation is $[1; 1, 1, \ldots, 1]$ which indeed is minimum.

When $q = 2$ each player either has weight 1 or 2. We can list all the possible representations, starting from the one in which all the players have weight 1, and increase the number of twos, until reaching the representation in which all the players have weight 2. In this sequence, all but the two representations $[2; 2, 2, \ldots, 2]$ and $[2; 2, 2, \ldots, 2, 1]$ are canonical minimum. Observe that the first excluded one does not comply the property $gcd(q, w_1, w_2, \ldots, w_n) = 1$ of Proposition 1. In the second one, player $n$ does not belong to any minimal winning coalition, and hence it is a dummy. Therefore the condition that all dummy players have weight 0 is violated. In total, we have $n + 1$ representations, two of them not being minimum sum, and therefore we have $n - 1$ distinct minimum sum representations.

When $q = 3$, in a minimum representation $\mathbf{w}$ can only hold weights 1, 2 and 3. The representation with all weights equal to 3 is not minimum, therefore at least one player must have weight 1 or 2. According to Proposition 2, a player with assigned weight 1 must be in a minimal winning coalition of weight 3. Therefore, there are two possibilities, either there are 3 distinct players with weight 1 or there is a player with weight 2. Note that if there is only one player with weight 1 and some players with weight 2 and 3, i.e., $[3; 3, \ldots, 3, 2, \ldots, 2, 1]$, the representation is not minimum sum, since $[2; 2, \ldots, 2, 1, \ldots, 1]$ has less sum and represents the same game. Note, however that if an extra player with weight 1 is added then the representation is minimum sum. It is easy to check that any combinations avoiding the mentioned restrictions are minimum sum. Therefore, we have only two types of weights assignments to consider. Assignments with at least three ones and assignments with at least a two and at least two ones. Counting the first type is equivalent to counting the number of integer solutions of the equation $x_1 + x_2 + x_3 = n$ with $x_2, x_3 \geq 0$ and $x_1 \geq 3$. Furthermore, the second one is equivalent to the number of integer solutions to the equation $x_1 + x_2 + x_3 = n$ with $x_1 \geq 2$, $x_2 \geq 1$ and $x_3 \geq 0$. In both cases the total number is $\binom{n-1}{2}$. However, we are double counting some solutions. We need to discount the number of integer solutions to the equation $x_1 + x_2 + x_3 = n$ with $x_1 \geq 3$, $x_2 \geq 1$ and $x_3 \geq 0$ which is $\binom{n-2}{2}$. Therefore, $M(n, 3) = 2\binom{n-1}{2} - \binom{n-2}{2} = \frac{(n-2)(n+1)}{2}$ as we wanted to see.  □

We can also get the following expressions for general games with quota up to 3.

**Proposition 8.** *For any $n > 1$, $D(n, 1) = n$, $D(n, 2) = \binom{n}{2}$, $D(n, 3) = \frac{(n-1)(n-2)(n+3)}{6}$.*

**Proof.** Recall that according to Proposition 1 dummy players in minimum sum representation have weight 0. When $q = 1$, we just need to select the number of players that are dummies. As the minimum number of dummies is 0, and the maximum is $n - 1$, we get a total of $n$ canonical minimum representations with quota 1.

When $q = 2$, now we have three possible weights: 0, 1 or 2. In order to get that the gcd of all values is 1, we need that at least one player has weight 1. However, if we only have one player with weight 1 this player would be a dummy. Therefore, we have, in any minimum sum representation, at least two players with weight 1. Furthermore, any

representation with at least 2 players with weight 1 is minimum sum. It can be trivially checked that none of its weights can be decreased. Therefore, $D(n, 2)$ is the number of non-negative integer solutions to the equation: $x_0 + x_1 + x_2 = n$, with $x_0, x_2, x_3 \geq 0$ and $x_1 \geq 2$. Which is indeed $\binom{n}{2}$.

Let us analyze the case $q = 3$. Deleting a dummy in a minimum sum representation of the game leads to the minimum sum representation of the game with the same minimal winning coalitions but with one player less. Therefore, conditions established for the minimum number of occurrences of the weights required in the proof of Proposition 7 must be preserved here. Therefore, to compute $D(n, 3)$ it suffices to compute the number of integral solutions to the equation $x_0 + x_1 + x_2 + x_3 = n$, first with $x_0, x_2, x_3 \geq 0$ and $x_1 \geq 3$; second with $x_0, x_3 \geq 0$, $x_1 \geq 2$ and $x_2 \geq 1$; third with $x_0, x_3 \geq 0$, $x_1 \geq 3$ and $x_2 \geq 1$. As before, the last number takes care of the representations that are counted in the other two. This gives, $D(n, 3) = \binom{n}{3} + \binom{n}{3} - \binom{n-1}{3} = \frac{(n-1)(n-2)(n+3)}{6}$.  □

Up to quota 3, we observe a polynomial number of representations, which agree with Proposition 6.

In the previous results, we focused in the number of WMVG allowing or not dummies. Those results can be used to count other subclasses of WMVG for small quotas. We call player *i* is *winner* in a game $\Gamma$ when the coalition $\{i\}$ is winning. Furthermore, the weight of a winner in a minimum sum representation determines the quota. In the proof of Proposition 4, we have shown that if $[q; \mathbf{w}]$ is a minimum representation of $\Gamma$ and we consider the game $\Gamma'$ removing a winner in $\Gamma$, the remaining weights with quota $q$ are a minimum representation for $\Gamma'$. Using this property we get the following result.

**Lemma 3.** *For $n > 1$, the number of minimum representations of WMVGs without dummies, with n players and quota q having a winner player is equal to the number of minimum representations of WMVGs without dummies with $n - 1$ players and quota q.*

## 7. Discussion and Conclusions

We have analyzed the weights and the quotas appearing in canonical minimum sum representations of WMVGs up to 8 players. Our analysis draws a clear picture of the frequency and distributions of such values. We have observed that the distributions become more similar as the number of players increases. The predicted distributions could help to find a method to obtain randomly canonical minimum representations of games with a large number of players. Furthermore, such a distribution might provide the tool to analyze other relevant question on the set of weighted voting games, in particular the relationship among weight and power.

We have devised some simple procedures that allow us to obtain extended minimum or minimum sum representations by the addition of one player to a minimum or minimum sum representation. A future line of work is to understand the size and properties of family of WMVGs that can be obtained through the proposed procedures. One of the procedures for minimum representations involve the repetition of one of the weights, i.e., $[q; w_i \oplus \mathbf{w}]$. For the case of minimum sum representations, in [29] it is shown that there are games having more than one minimum sum representation in which equivalent players get different weights. Assume that such equivalent players are *i* and *j* in a minimum sum representation $[q; \mathbf{w}]$. Then $[q + w_i; w_i \oplus \mathbf{w}]$ and $[q + w_j; w_j \oplus \mathbf{w}]$ represent the same game, and both cannot be minimum sum. Thus, this procedure does not allow to create an extended minimum sum representation. It remains open to show if the procedures $[q + w_i; w_i \oplus \mathbf{w}]$ or $[q; q \oplus \mathbf{w}]$ are valid for minimum sum representations.

One consequence of these procedures is that, for any quota or weight, it is possible to generate a WMVG with minimum or minimum sum representation containing this weight or quota for a large enough number of players. In this line, we have some open problems related to the considered values in Corollary 1. Firstly, given a quota $q \geq 1$, to find the minimum number of players $n_q$ such that there exits a WMVG with minimum (sum)

representation $[q; \mathbf{w}]$. In the same vein, given a weight $w \geq 0$, to find the minimum number of players $n_w$ such that there exits a WMVG with minimum (sum) representation $[q; w \oplus \mathbf{w}]$.

As a consequence of the previous procedures, we have obtained bounds among the maximum weights and quotas in minimum representations. However the maximum values coincide up to eight players. An interesting problem is to determine whether the relationships carry over to the maximum values in canonical minimum representations.

We have proved that, for a quota $q \leq 3$, all minimum sum representations are minimum. However, the representation $[12; 7, 6, 6, 4, 4, 4, 3, 2]$ given in [29] is a minimum sum representation, but it is not a minimum representation. It is interesting to find the smallest quota $3 < q < 12$ such that there exists a WMVG without minimum representation.

It remains open to find closed formulas for $M(n, q)$ and $D(n, q)$, when we restrict ourselves to subclasses of WMVGs as, for example, self-dual or non seft-dual.

Our last results provide information on games with multiple players having canonical minimum representations with small quotas. Those games allow for a simpler representation in which we need only to state the number of times that each weight appears. This representation might lead to fast algorithms for listing or enumerating the canonical minimum representations of games with many players and a reasonable small quota.

**Author Contributions:** Conceptualization, X.M., M.S. and M.T.-O.; methodology, X.M., M.S. and M.T.-O.; investigation, X.M., M.S. and M.T.-O.; writing—original draft preparation, X.M., M.S. and M.T.-O.; writing—review and editing, X.M., M.S. and M.T.-O. All authors have read and agreed to the published version of the manuscript.

**Funding:** The research of X. Molinero has been partially supported by funds from the Spanish Agencia Estatal de Investigación under grant PID2019-104987GB-I00 (JUVOCO). M. Serna was partially supported by funds from the Spanish Agencia Estatal de Investigación under grant PID2020-112581GB-C21 (MOTION) and from the Catalan Agència de Gestió d'Ajuts Universitaris i de Recerca (Agaur) under project ALBCOM 2017-SGR-786.

**Institutional Review Board Statement:** Not applicable.

**Informed Consent Statement:** Not applicable.

**Data Availability Statement:** Data was obtained from Xavier Molinero and it is available on request.

**Acknowledgments:** We thank the anonymous referees for their careful reading and helpful suggestions.

**Conflicts of Interest:** The authors declare no conflict of interest.

## Abbreviations

The following abbreviations are used in this manuscript:

| | |
|---|---|
| SG | Simple Game |
| WMVG | Weighted Majority Voting Game |
| $\Gamma^d$ | dual of $\Gamma$ |
| $\mathcal{L}$ | set of losing coalitions |
| $\mathcal{L}^M$ | set of maximal losing coalition |
| $\mathcal{W}^m$ | set of minimal winning coalitions |
| $\mathcal{W}$ | set of winning coalitions |
| $\mathcal{W}^d$ | set of winning coalitions of the dual game |
| $q_n^{\max}$ | maximum quota in a canonical minimum representation |
| $\overline{q}_n^{\max}$ | maximum quota in a minimum representation |
| $w_n^{\max}$ | maximum weight in a canonical minimum representation |
| $\overline{w}_n^{\max}$ | maximum weight in a minimum representation |
| $w_n^{\text{u-min}}$ | minimum non-repeated weight in a canonical minimum representation |
| $\overline{w}_n^{\text{u-min}}$ | minimum non-repeated weight in a minimum representation |

## Appendix A

In this section, we incorporate all the data gathered about the frequencies of weights and quotas for games up to 8 players.

*Appendix A.1*

**Table A1.** Frequencies of weights for canonical minimum game representations.

| Weight | # of Players | | | | | | | | Total |
|---|---|---|---|---|---|---|---|---|---|
| | 1 | 2 | 3 | 4 | 5 | 6 | 7 | 8 | |
| 1 | 1 | 4 | 13 | 45 | 196 | 1349 | 20,288 | 933,039 | 954,935 |
| 2 | 0 | 0 | 2 | 17 | 134 | 1416 | 28,148 | 1,513,774 | 1,543,491 |
| 3 | 0 | 0 | 0 | 6 | 82 | 1144 | 26,702 | 1,602,456 | 1,630,390 |
| 4 | 0 | 0 | 0 | 0 | 30 | 744 | 23,376 | 1,599,991 | 1,624,141 |
| 5 | 0 | 0 | 0 | 0 | 18 | 607 | 21,487 | 1,543,328 | 1,565,440 |
| 6 | 0 | 0 | 0 | 0 | 0 | 298 | 16,826 | 1,465,011 | 1,482,135 |
| 7 | 0 | 0 | 0 | 0 | 0 | 238 | 15,211 | 1,397,070 | 1,412,519 |
| 8 | 0 | 0 | 0 | 0 | 0 | 110 | 11,592 | 1,295,818 | 1,307,520 |
| 9 | 0 | 0 | 0 | 0 | 0 | 58 | 9768 | 1,212,111 | 1,221,937 |
| 10 | 0 | 0 | 0 | 0 | 0 | 0 | 6872 | 1,103,819 | 1,110,691 |
| 11 | 0 | 0 | 0 | 0 | 0 | 0 | 5972 | 1,032,565 | 1,038,537 |
| 12 | 0 | 0 | 0 | 0 | 0 | 0 | 4036 | 920,263 | 924,299 |
| 13 | 0 | 0 | 0 | 0 | 0 | 0 | 3262 | 847,566 | 850,828 |
| 14 | 0 | 0 | 0 | 0 | 0 | 0 | 1932 | 746,821 | 748,753 |
| 15 | 0 | 0 | 0 | 0 | 0 | 0 | 1158 | 667,047 | 668,205 |
| 16 | 0 | 0 | 0 | 0 | 0 | 0 | 724 | 595,577 | 596,301 |
| 17 | 0 | 0 | 0 | 0 | 0 | 0 | 298 | 522,479 | 522,777 |
| 18 | 0 | 0 | 0 | 0 | 0 | 0 | 182 | 459,325 | 459,507 |
| 19 | 0 | 0 | 0 | 0 | 0 | 0 | 0 | 395,566 | 395,566 |
| 20 | 0 | 0 | 0 | 0 | 0 | 0 | 0 | 343,714 | 343,714 |
| 21 | 0 | 0 | 0 | 0 | 0 | 0 | 0 | 285,876 | 285,876 |
| 22 | 0 | 0 | 0 | 0 | 0 | 0 | 0 | 244,044 | 244,044 |
| 23 | 0 | 0 | 0 | 0 | 0 | 0 | 0 | 195,572 | 195,572 |
| 24 | 0 | 0 | 0 | 0 | 0 | 0 | 0 | 170,640 | 170,640 |
| 25 | 0 | 0 | 0 | 0 | 0 | 0 | 0 | 121,872 | 121,872 |
| 26 | 0 | 0 | 0 | 0 | 0 | 0 | 0 | 105,500 | 105,500 |
| 27 | 0 | 0 | 0 | 0 | 0 | 0 | 0 | 73,660 | 73,660 |
| 28 | 0 | 0 | 0 | 0 | 0 | 0 | 0 | 66,696 | 66,696 |
| 29 | 0 | 0 | 0 | 0 | 0 | 0 | 0 | 37,858 | 37,858 |
| 30 | 0 | 0 | 0 | 0 | 0 | 0 | 0 | 38,588 | 38,588 |
| 31 | 0 | 0 | 0 | 0 | 0 | 0 | 0 | 19,946 | 19,946 |
| 32 | 0 | 0 | 0 | 0 | 0 | 0 | 0 | 16,158 | 16,158 |
| 33 | 0 | 0 | 0 | 0 | 0 | 0 | 0 | 9894 | 9894 |
| 34 | 0 | 0 | 0 | 0 | 0 | 0 | 0 | 11,020 | 11,020 |
| 35 | 0 | 0 | 0 | 0 | 0 | 0 | 0 | 3632 | 3632 |
| 36 | 0 | 0 | 0 | 0 | 0 | 0 | 0 | 3312 | 3312 |
| 37 | 0 | 0 | 0 | 0 | 0 | 0 | 0 | 672 | 672 |
| 38 | 0 | 0 | 0 | 0 | 0 | 0 | 0 | 2656 | 2656 |
| 39 | 0 | 0 | 0 | 0 | 0 | 0 | 0 | 208 | 208 |
| 40 | 0 | 0 | 0 | 0 | 0 | 0 | 0 | 992 | 992 |
| 41 | 0 | 0 | 0 | 0 | 0 | 0 | 0 | 0 | 0 |
| 42 | 0 | 0 | 0 | 0 | 0 | 0 | 0 | 192 | 192 |
| Total | 1 | 4 | 15 | 68 | 460 | 5964 | 197,834 | 21,606,328 | 21,810,674 |

*Appendix A.2*

**Table A2.** Frequencies of weights in strict canonical minimum representations.

| Weight | # of Players | | | | | | | | Total |
|---|---|---|---|---|---|---|---|---|---|
| | **1** | **2** | **3** | **4** | **5** | **6** | **7** | **8** | |
| 1 | 1 | 2 | 5 | 15 | 59 | 375 | 5315 | 237,538 | 243,310 |
| 2 | 0 | 0 | 1 | 6 | 41 | 389 | 7277 | 386,471 | 394,185 |
| 3 | 0 | 0 | 0 | 3 | 29 | 341 | 7068 | 412,834 | 420,275 |
| 4 | 0 | 0 | 0 | 0 | 12 | 226 | 6260 | 411,586 | 418,084 |
| 5 | 0 | 0 | 0 | 0 | 9 | 214 | 6126 | 404,808 | 411,157 |
| 6 | 0 | 0 | 0 | 0 | 0 | 102 | 4709 | 382,773 | 387,584 |
| 7 | 0 | 0 | 0 | 0 | 0 | 101 | 4709 | 378,859 | 383,669 |
| 8 | 0 | 0 | 0 | 0 | 0 | 47 | 3602 | 350,039 | 353,688 |
| 9 | 0 | 0 | 0 | 0 | 0 | 29 | 3245 | 337,054 | 340,328 |
| 10 | 0 | 0 | 0 | 0 | 0 | 0 | 2303 | 310,445 | 312,748 |
| 11 | 0 | 0 | 0 | 0 | 0 | 0 | 2224 | 302,672 | 304,896 |
| 12 | 0 | 0 | 0 | 0 | 0 | 0 | 1495 | 267,925 | 269,420 |
| 13 | 0 | 0 | 0 | 0 | 0 | 0 | 1400 | 259,825 | 261,225 |
| 14 | 0 | 0 | 0 | 0 | 0 | 0 | 825 | 229,211 | 230,036 |
| 15 | 0 | 0 | 0 | 0 | 0 | 0 | 543 | 212,100 | 212,643 |
| 16 | 0 | 0 | 0 | 0 | 0 | 0 | 318 | 191,634 | 191,952 |
| 17 | 0 | 0 | 0 | 0 | 0 | 0 | 149 | 176,428 | 176,577 |
| 18 | 0 | 0 | 0 | 0 | 0 | 0 | 91 | 156,578 | 156,669 |
| 19 | 0 | 0 | 0 | 0 | 0 | 0 | 0 | 143,393 | 143,393 |
| 20 | 0 | 0 | 0 | 0 | 0 | 0 | 0 | 124,447 | 124,447 |
| 21 | 0 | 0 | 0 | 0 | 0 | 0 | 0 | 110,904 | 110,904 |
| 22 | 0 | 0 | 0 | 0 | 0 | 0 | 0 | 94,874 | 94,874 |
| 23 | 0 | 0 | 0 | 0 | 0 | 0 | 0 | 82,068 | 82,068 |
| 24 | 0 | 0 | 0 | 0 | 0 | 0 | 0 | 69,547 | 69,547 |
| 25 | 0 | 0 | 0 | 0 | 0 | 0 | 0 | 54,208 | 54,208 |
| 26 | 0 | 0 | 0 | 0 | 0 | 0 | 0 | 46,049 | 46,049 |
| 27 | 0 | 0 | 0 | 0 | 0 | 0 | 0 | 34,135 | 34,135 |
| 28 | 0 | 0 | 0 | 0 | 0 | 0 | 0 | 30,404 | 30,404 |
| 29 | 0 | 0 | 0 | 0 | 0 | 0 | 0 | 18,216 | 18,216 |
| 30 | 0 | 0 | 0 | 0 | 0 | 0 | 0 | 18,355 | 18,355 |
| 31 | 0 | 0 | 0 | 0 | 0 | 0 | 0 | 9762 | 9762 |
| 32 | 0 | 0 | 0 | 0 | 0 | 0 | 0 | 7818 | 7818 |
| 33 | 0 | 0 | 0 | 0 | 0 | 0 | 0 | 4894 | 4894 |
| 34 | 0 | 0 | 0 | 0 | 0 | 0 | 0 | 5442 | 5442 |
| 35 | 0 | 0 | 0 | 0 | 0 | 0 | 0 | 1816 | 1816 |
| 36 | 0 | 0 | 0 | 0 | 0 | 0 | 0 | 1656 | 1656 |
| 37 | 0 | 0 | 0 | 0 | 0 | 0 | 0 | 336 | 336 |
| 38 | 0 | 0 | 0 | 0 | 0 | 0 | 0 | 1328 | 1328 |
| 39 | 0 | 0 | 0 | 0 | 0 | 0 | 0 | 104 | 104 |
| 40 | 0 | 0 | 0 | 0 | 0 | 0 | 0 | 496 | 496 |
| 41 | 0 | 0 | 0 | 0 | 0 | 0 | 0 | 0 | 0 |
| 42 | 0 | 0 | 0 | 0 | 0 | 0 | 0 | 96 | 96 |
| Total | 1 | 2 | 6 | 24 | 150 | 1824 | 57,659 | 6,269,128 | 6,328,794 |

*Appendix A.3*

**Table A3.** Frequencies of weights in canonical minimum representations disregarding multiplicities.

| Weight | # of Players | | | | | | | | Total |
|---|---|---|---|---|---|---|---|---|---|
| | **1** | **2** | **3** | **4** | **5** | **6** | **7** | **8** | |
| 1 | 1 | 2 | 5 | 17 | 86 | 760 | 14,751 | 807,905 | 823,527 |
| 2 | 0 | 0 | 2 | 11 | 72 | 794 | 18,901 | 1,225,441 | 1,245,221 |
| 3 | 0 | 0 | 0 | 6 | 58 | 721 | 18,334 | 1,286,604 | 1,305,723 |
| 4 | 0 | 0 | 0 | 0 | 30 | 570 | 17,156 | 1,295,455 | 1,313,211 |
| 5 | 0 | 0 | 0 | 0 | 18 | 497 | 16,383 | 1,262,061 | 1,278,959 |
| 6 | 0 | 0 | 0 | 0 | 0 | 298 | 14,006 | 1,225,101 | 1,239,405 |
| 7 | 0 | 0 | 0 | 0 | 0 | 238 | 13,047 | 1,183,799 | 1,197,084 |
| 8 | 0 | 0 | 0 | 0 | 0 | 110 | 10,530 | 1,123,504 | 1,134,144 |
| 9 | 0 | 0 | 0 | 0 | 0 | 58 | 9212 | 1,069,176 | 1,078,446 |
| 10 | 0 | 0 | 0 | 0 | 0 | 0 | 6872 | 996,529 | 1,003,401 |
| 11 | 0 | 0 | 0 | 0 | 0 | 0 | 5972 | 943,919 | 949,891 |
| 12 | 0 | 0 | 0 | 0 | 0 | 0 | 4036 | 857,467 | 861,503 |
| 13 | 0 | 0 | 0 | 0 | 0 | 0 | 3262 | 799,922 | 803,184 |
| 14 | 0 | 0 | 0 | 0 | 0 | 0 | 1932 | 717,731 | 719,663 |
| 15 | 0 | 0 | 0 | 0 | 0 | 0 | 1158 | 649,807 | 650,965 |
| 16 | 0 | 0 | 0 | 0 | 0 | 0 | 724 | 584,161 | 584,885 |
| 17 | 0 | 0 | 0 | 0 | 0 | 0 | 298 | 518,361 | 518,659 |
| 18 | 0 | 0 | 0 | 0 | 0 | 0 | 182 | 456,343 | 456,525 |
| 19 | 0 | 0 | 0 | 0 | 0 | 0 | 0 | 395,566 | 395,566 |
| 20 | 0 | 0 | 0 | 0 | 0 | 0 | 0 | 343,714 | 343,714 |
| 21 | 0 | 0 | 0 | 0 | 0 | 0 | 0 | 285,876 | 285,876 |
| 22 | 0 | 0 | 0 | 0 | 0 | 0 | 0 | 244,044 | 244,044 |
| 23 | 0 | 0 | 0 | 0 | 0 | 0 | 0 | 195,572 | 195,572 |
| 24 | 0 | 0 | 0 | 0 | 0 | 0 | 0 | 170,640 | 170,640 |
| 25 | 0 | 0 | 0 | 0 | 0 | 0 | 0 | 121,872 | 121,872 |
| 26 | 0 | 0 | 0 | 0 | 0 | 0 | 0 | 105,500 | 105,500 |
| 27 | 0 | 0 | 0 | 0 | 0 | 0 | 0 | 73,660 | 73,660 |
| 28 | 0 | 0 | 0 | 0 | 0 | 0 | 0 | 66,696 | 66,696 |
| 29 | 0 | 0 | 0 | 0 | 0 | 0 | 0 | 37,858 | 37,858 |
| 30 | 0 | 0 | 0 | 0 | 0 | 0 | 0 | 38,588 | 38,588 |
| 31 | 0 | 0 | 0 | 0 | 0 | 0 | 0 | 19,946 | 19,946 |
| 32 | 0 | 0 | 0 | 0 | 0 | 0 | 0 | 16,158 | 16,158 |
| 33 | 0 | 0 | 0 | 0 | 0 | 0 | 0 | 9894 | 9894 |
| 34 | 0 | 0 | 0 | 0 | 0 | 0 | 0 | 11,020 | 11,020 |
| 35 | 0 | 0 | 0 | 0 | 0 | 0 | 0 | 3632 | 3632 |
| 36 | 0 | 0 | 0 | 0 | 0 | 0 | 0 | 3312 | 3312 |
| 37 | 0 | 0 | 0 | 0 | 0 | 0 | 0 | 672 | 672 |
| 38 | 0 | 0 | 0 | 0 | 0 | 0 | 0 | 2656 | 2656 |
| 39 | 0 | 0 | 0 | 0 | 0 | 0 | 0 | 208 | 208 |
| 40 | 0 | 0 | 0 | 0 | 0 | 0 | 0 | 992 | 992 |
| 41 | 0 | 0 | 0 | 0 | 0 | 0 | 0 | 0 | 0 |
| 42 | 0 | 0 | 0 | 0 | 0 | 0 | 0 | 192 | 192 |
| Total | 1 | 2 | 7 | 34 | 264 | 4046 | 156,756 | 19,151,554 | 19,312,664 |

*Appendix A.4*

**Table A4.** Frequencies of quotas in canonical minimum representations.

| Quota | # of Players | | | | | | | | Total |
|---|---|---|---|---|---|---|---|---|---|
| | **1** | **2** | **3** | **4** | **5** | **6** | **7** | **8** | |
| 1 | 1 | 1 | 1 | 1 | 1 | 1 | 1 | 1 | 8 |
| 2 | 0 | 1 | 2 | 3 | 4 | 5 | 6 | 7 | 28 |
| 3 | 0 | 0 | 2 | 5 | 9 | 14 | 20 | 27 | 77 |
| 4 | 0 | 0 | 0 | 5 | 15 | 31 | 54 | 85 | 190 |
| 5 | 0 | 0 | 0 | 3 | 17 | 47 | 100 | 184 | 351 |
| 6 | 0 | 0 | 0 | 0 | 16 | 72 | 195 | 421 | 704 |
| 7 | 0 | 0 | 0 | 0 | 16 | 88 | 288 | 720 | 1112 |
| 8 | 0 | 0 | 0 | 0 | 9 | 101 | 429 | 1267 | 1806 |
| 9 | 0 | 0 | 0 | 0 | 5 | 109 | 577 | 1963 | 2654 |
| 10 | 0 | 0 | 0 | 0 | 0 | 108 | 769 | 3066 | 3943 |
| 11 | 0 | 0 | 0 | 0 | 0 | 114 | 947 | 4258 | 5319 |
| 12 | 0 | 0 | 0 | 0 | 0 | 81 | 1087 | 5999 | 7167 |
| 13 | 0 | 0 | 0 | 0 | 0 | 89 | 1310 | 7971 | 9370 |
| 14 | 0 | 0 | 0 | 0 | 0 | 55 | 1432 | 10,452 | 11,939 |
| 15 | 0 | 0 | 0 | 0 | 0 | 40 | 1557 | 13,119 | 14,716 |
| 16 | 0 | 0 | 0 | 0 | 0 | 15 | 1604 | 16,381 | 18,000 |
| 17 | 0 | 0 | 0 | 0 | 0 | 17 | 1794 | 20,070 | 21,881 |
| 18 | 0 | 0 | 0 | 0 | 0 | 7 | 1700 | 23,746 | 25,453 |
| 19 | 0 | 0 | 0 | 0 | 0 | 0 | 1828 | 28,328 | 30,156 |
| 20 | 0 | 0 | 0 | 0 | 0 | 0 | 1682 | 32,403 | 34,085 |
| 21 | 0 | 0 | 0 | 0 | 0 | 0 | 1661 | 37,203 | 38,864 |
| 22 | 0 | 0 | 0 | 0 | 0 | 0 | 1413 | 41,463 | 42,876 |
| 23 | 0 | 0 | 0 | 0 | 0 | 0 | 1504 | 47,652 | 49,156 |
| 24 | 0 | 0 | 0 | 0 | 0 | 0 | 1168 | 50,625 | 51,793 |
| 25 | 0 | 0 | 0 | 0 | 0 | 0 | 1134 | 57,212 | 58,346 |
| 26 | 0 | 0 | 0 | 0 | 0 | 0 | 823 | 60,451 | 61,274 |
| 27 | 0 | 0 | 0 | 0 | 0 | 0 | 774 | 66,225 | 66,999 |
| 28 | 0 | 0 | 0 | 0 | 0 | 0 | 594 | 68,945 | 69,539 |
| 29 | 0 | 0 | 0 | 0 | 0 | 0 | 485 | 75,531 | 76,016 |
| 30 | 0 | 0 | 0 | 0 | 0 | 0 | 412 | 76,086 | 76,498 |
| 31 | 0 | 0 | 0 | 0 | 0 | 0 | 281 | 82,142 | 82,423 |
| 32 | 0 | 0 | 0 | 0 | 0 | 0 | 148 | 82,507 | 82,655 |
| 33 | 0 | 0 | 0 | 0 | 0 | 0 | 165 | 87,052 | 87,217 |
| 34 | 0 | 0 | 0 | 0 | 0 | 0 | 148 | 85,949 | 86,097 |
| 35 | 0 | 0 | 0 | 0 | 0 | 0 | 67 | 90,623 | 90,690 |
| 36 | 0 | 0 | 0 | 0 | 0 | 0 | 48 | 86,982 | 87,030 |
| 37 | 0 | 0 | 0 | 0 | 0 | 0 | 16 | 90,458 | 90,474 |
| 38 | 0 | 0 | 0 | 0 | 0 | 0 | 25 | 86,963 | 86,988 |
| 39 | 0 | 0 | 0 | 0 | 0 | 0 | 8 | 88,791 | 88,799 |
| 40 | 0 | 0 | 0 | 0 | 0 | 0 | 8 | 82,946 | 82,954 |
| 41 | 0 | 0 | 0 | 0 | 0 | 0 | 0 | 84,557 | 84,557 |
| 42 | 0 | 0 | 0 | 0 | 0 | 0 | 0 | 78,669 | 78,669 |
| 43 | 0 | 0 | 0 | 0 | 0 | 0 | 0 | 78,632 | 78,632 |
| 44 | 0 | 0 | 0 | 0 | 0 | 0 | 0 | 72,350 | 72,350 |
| 45 | 0 | 0 | 0 | 0 | 0 | 0 | 0 | 71,709 | 71,709 |
| 46 | 0 | 0 | 0 | 0 | 0 | 0 | 0 | 64,460 | 64,460 |
| 47 | 0 | 0 | 0 | 0 | 0 | 0 | 0 | 62,589 | 62,589 |
| 48 | 0 | 0 | 0 | 0 | 0 | 0 | 0 | 57,110 | 57,110 |
| 49 | 0 | 0 | 0 | 0 | 0 | 0 | 0 | 54,556 | 54,556 |

**Table A4.** *Cont.*

| Quota | # of Players | | | | | | | | Total |
|---|---|---|---|---|---|---|---|---|---|
| | 1 | 2 | 3 | 4 | 5 | 6 | 7 | 8 | |
| 50 | 0 | 0 | 0 | 0 | 0 | 0 | 0 | 49,433 | 49,433 |
| 51 | 0 | 0 | 0 | 0 | 0 | 0 | 0 | 47,505 | 47,505 |
| 52 | 0 | 0 | 0 | 0 | 0 | 0 | 0 | 41,084 | 41,084 |
| 53 | 0 | 0 | 0 | 0 | 0 | 0 | 0 | 36,881 | 36,881 |
| 54 | 0 | 0 | 0 | 0 | 0 | 0 | 0 | 35,016 | 35,016 |
| 55 | 0 | 0 | 0 | 0 | 0 | 0 | 0 | 32,361 | 32,361 |
| 56 | 0 | 0 | 0 | 0 | 0 | 0 | 0 | 26,769 | 26,769 |
| 57 | 0 | 0 | 0 | 0 | 0 | 0 | 0 | 24,945 | 24,945 |
| 58 | 0 | 0 | 0 | 0 | 0 | 0 | 0 | 23,499 | 23,499 |
| 59 | 0 | 0 | 0 | 0 | 0 | 0 | 0 | 19,829 | 19,829 |
| 60 | 0 | 0 | 0 | 0 | 0 | 0 | 0 | 16,793 | 16,793 |
| 61 | 0 | 0 | 0 | 0 | 0 | 0 | 0 | 15,265 | 15,265 |
| 62 | 0 | 0 | 0 | 0 | 0 | 0 | 0 | 13,142 | 13,142 |
| 63 | 0 | 0 | 0 | 0 | 0 | 0 | 0 | 11,201 | 11,201 |
| 64 | 0 | 0 | 0 | 0 | 0 | 0 | 0 | 10,612 | 10,612 |
| 65 | 0 | 0 | 0 | 0 | 0 | 0 | 0 | 8872 | 8872 |
| 66 | 0 | 0 | 0 | 0 | 0 | 0 | 0 | 8018 | 8018 |
| 67 | 0 | 0 | 0 | 0 | 0 | 0 | 0 | 6544 | 6544 |
| 68 | 0 | 0 | 0 | 0 | 0 | 0 | 0 | 4874 | 4874 |
| 69 | 0 | 0 | 0 | 0 | 0 | 0 | 0 | 4446 | 4446 |
| 70 | 0 | 0 | 0 | 0 | 0 | 0 | 0 | 4020 | 4020 |
| 71 | 0 | 0 | 0 | 0 | 0 | 0 | 0 | 3031 | 3031 |
| 72 | 0 | 0 | 0 | 0 | 0 | 0 | 0 | 2957 | 2957 |
| 73 | 0 | 0 | 0 | 0 | 0 | 0 | 0 | 2563 | 2563 |
| 74 | 0 | 0 | 0 | 0 | 0 | 0 | 0 | 1776 | 1776 |
| 75 | 0 | 0 | 0 | 0 | 0 | 0 | 0 | 1820 | 1820 |
| 76 | 0 | 0 | 0 | 0 | 0 | 0 | 0 | 1283 | 1283 |
| 77 | 0 | 0 | 0 | 0 | 0 | 0 | 0 | 820 | 820 |
| 78 | 0 | 0 | 0 | 0 | 0 | 0 | 0 | 770 | 770 |
| 79 | 0 | 0 | 0 | 0 | 0 | 0 | 0 | 900 | 900 |
| 80 | 0 | 0 | 0 | 0 | 0 | 0 | 0 | 533 | 533 |
| 81 | 0 | 0 | 0 | 0 | 0 | 0 | 0 | 418 | 418 |
| 82 | 0 | 0 | 0 | 0 | 0 | 0 | 0 | 481 | 481 |
| 83 | 0 | 0 | 0 | 0 | 0 | 0 | 0 | 332 | 332 |
| 84 | 0 | 0 | 0 | 0 | 0 | 0 | 0 | 215 | 215 |
| 85 | 0 | 0 | 0 | 0 | 0 | 0 | 0 | 225 | 225 |
| 86 | 0 | 0 | 0 | 0 | 0 | 0 | 0 | 143 | 143 |
| 87 | 0 | 0 | 0 | 0 | 0 | 0 | 0 | 43 | 43 |
| 88 | 0 | 0 | 0 | 0 | 0 | 0 | 0 | 79 | 79 |
| 89 | 0 | 0 | 0 | 0 | 0 | 0 | 0 | 58 | 58 |
| 90 | 0 | 0 | 0 | 0 | 0 | 0 | 0 | 33 | 33 |
| 91 | 0 | 0 | 0 | 0 | 0 | 0 | 0 | 110 | 110 |
| 92 | 0 | 0 | 0 | 0 | 0 | 0 | 0 | 56 | 56 |
| 93 | 0 | 0 | 0 | 0 | 0 | 0 | 0 | 20 | 20 |
| 94 | 0 | 0 | 0 | 0 | 0 | 0 | 0 | 37 | 37 |
| 95 | 0 | 0 | 0 | 0 | 0 | 0 | 0 | 0 | 0 |
| 96 | 0 | 0 | 0 | 0 | 0 | 0 | 0 | 45 | 45 |
| 97 | 0 | 0 | 0 | 0 | 0 | 0 | 0 | 0 | 0 |
| 98 | 0 | 0 | 0 | 0 | 0 | 0 | 0 | 1 | 1 |
| 99 | 0 | 0 | 0 | 0 | 0 | 0 | 0 | 0 | 0 |

**Table A4.** *Cont.*

| Quota | # of Players | | | | | | | | Total |
|---|---|---|---|---|---|---|---|---|---|
| | 1 | 2 | 3 | 4 | 5 | 6 | 7 | 8 | |
| 100 | 0 | 0 | 0 | 0 | 0 | 0 | 0 | 0 | 0 |
| 101 | 0 | 0 | 0 | 0 | 0 | 0 | 0 | 0 | 0 |
| 102 | 0 | 0 | 0 | 0 | 0 | 0 | 0 | 9 | 9 |
| 103 | 0 | 0 | 0 | 0 | 0 | 0 | 0 | 0 | 0 |
| 104 | 0 | 0 | 0 | 0 | 0 | 0 | 0 | 0 | 0 |
| 105 | 0 | 0 | 0 | 0 | 0 | 0 | 0 | 18 | 18 |
| Total | 1 | 2 | 5 | 17 | 92 | 994 | 28,262 | 270,0791 | 273,0164 |

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
