# Peer review of "On Weights and Quotas for Weighted Majority Voting Games"

_games, doi:10.3390/g12040091_

Round 1

Reviewer 1 Report

The authors discuss various issues related to WMVG, especially for a large number of players. Before I come to my comments, I would like to emphasize that I am not an expert in this field. In this sense, I would have liked to see more economic intuition, especially in the introduction. For example, why is it important for economists to know that weight 41 does not appear in any canonical minimal representation of games with 8 players? Perhaps it is possible to include 1-2 sentences on this topic.

Other than that, I have a few minor points:

- 114: typo "that"

- 151-152: sentence is odd

- 168: Is it really 10,000?

- 171 onwards: When comparing representations with more than (e.g. 4) players, it might be helpful to highlight the differences between the representations somehow (e.g. thick).

- Figure 1 onwards: It would be nice if the panels were the same size (especially Figure 6).

- 259: Is "Gaussian kernel" relevant to your results?

- 343-350: There are some typos in the proof of Proposition 4: "However, but ...", "exists", "like this with lead"

- 360: Skip the round brackets

- 377: Typo "that"

- 386: Sentence beginning with "Let..." is odd

- 404: Typo "an"

- 414: typo "there"

- 529: I would consistently use either "weight 1" or "weight one"

- 594: Typo: "in a minimum number of games"

Author Response

Point 1: ... In this sense, I would have liked to see more economic intuition, especially in the introduction. For example, why is it important for economists to know that weight 41 does not appear in any canonical minimal representation of games with 8 players? Perhaps it is possible to include 1-2 sentences on this topic.

Response 1: We have added some economic applications with references.About the skipped number 41 is only a characterization of canonical representations. It let us to add some future work to analyse why it happens, as well as, to study what happens with skipped weights when we consider canonical representations with a large number of players.

Point 2: ... I have a few minor points.

Response 2: Thanks you for you corrections. We have checked all of them and more.

Reviewer 2 Report

Dear Authors,

The considerations in your article relate to the analysis of weight distributions and quota sizes in simple games (weighted majority voting  games (WMVG)) with up to 8 players. Constructions leading to obtaining WMVG representation with minimal or minimal sum of weights were presented. The issues are interesting, and the results are used in voting systems. The introduction is clearly written, but the research motivation is not sufficiently motivated. In particular, it is not known whether the results could be useful for analyzing a class of weighted voting games in which the weights are randomly distributed with respect to the standard probability simplex (see, e.g., Boratyn, D. et al. (2020) Mean weights and power in weighted voting games. Math. Social Sci. 108 (2020), 90–99.). 

Therefore, I am asking for in-depth motivation, perhaps related to the technical applications of voting systems for the construction of expert systems, automation of alarm signals in supervision systems.

The work should be published after author's re-edition and modifications. 

Author Response

Point 1: The introduction is clearly written, but the research motivation is not sufficiently motivated. In particular, it is not known whether the results could be useful for analyzing a class of weighted voting games in which the weights are randomly distributed with respect to the standard probability simplex (see, e.g., Boratyn, D. et al. (2020) Mean weights and power in weighted voting games. Math. Social Sci. 108 (2020), 90–99.). 

Response 1: We have added some sentences (with references) to motivate the paper according this comment/suggestion. In particular, we have motivated about randomly distributed weights.

Point 2: Therefore, I am asking for in-depth motivation, perhaps related to the technical applications of voting systems for the construction of expert systems, automation of alarm signals in supervision systems.

Various application in decision theory of voting systems can be found e.g. Szajowski, K. and Yasuda, M. (1997) Voting procedure on stopping games of Markov chain. (English summary) Stochastic modelling in innovative manufacturing (Cambridge, 1995), 68–80,
Lecture Notes in Econom. and Math. Systems, 445, Springer, Berlin, 1997. 

Response 2: In this line, we have added some references about applications in decision theory systems.

Reviewer 3 Report

In my view, the paper is an excellent contribution to this special issue. It presents several related but separate investigations that I consider substantial and very competently executed. The findings on how minimum, minimum sum, or canonical minimum sum representations look like, how they can be extended from a small to a larger set of players, etc. are of clear interest to specialist scholars and more generally improve our understanding of weighted majority voting. I only have minor presentational issues, which I list below. I can happily recommend acceptance of the paper subject to a little polishing.

Suggestions (include various typo corrections; better double-check for more):

line 13: “i.e.”

line 49: “a lot of work has been devoted”

line 80/81 and several other places: The terms “(dis)continuous” and “(dis)contiguous” are both used in the paper to refer to gaps in a discrete set. I would prefer to adopt “(dis)contiguous” throughout. Saying “the range of weight values or quotas is continuous” rings the wrong analysis/set theoretical bells, at least for me.

line 85f: “with many players”

line 107, 109: “players’ weights”

line 161f: please double-check the grammar of this sentence

line 226: “but that never appears more”

line 251: “as the number of players increases, so does”

Figure 1: “weights in canonical”

line 258ff, line 318ff: I am unfamiliar with this kind of density estimation and the interpretation of Figures 2 and 6. What is on the x-axis? What use is the density estimation? (How) does it relate to the suggestion “it might be possible to get good representations from the distribution of weights and quotas in the target representations” in the Introduction? I would think it worthwhile to explain and motivate the allegedly “standard probabilistic approach” (line 256) to readers having a background that is different from the authors.

line 268: is a “strict” missing here?

line 290: sentence seems to contain too many commas

line 311: “carries over to” ?

Figure 5: “Distribution”

line 327: “computationally hard”

The proofs of Prop. 4, Prop. 5 and Corollary 1 all contain too many typos for my taste. Please double-check them carefully. For instance: line 344f “However, all”; line 344 “become”; line 346 “From Proposition 1.1” (isn’t it?); line 348 “that there exists” etc. etc.

line 373: “lead to a different”

line 377: “smaller than”

line 379: “constructed by”

line 427: “minimum representations”

line 501f: The grammar of the sentence is not clear to me.

line 576: “determines”

line 588: It is not clear to me which notion of “statistical significance” is employed or referred to here. (I had the same problem already in line 78 but thought this would become clear later, which it didn’t.) What is the hypothesis and which test is involved? What is the relevant test statistic and which level of confidence is considered? Please either clarify or drop this point.

line 608: “having more than”

line 623: “crossed relationships” isn’t clear to me

line 626: “relationships extend to the maximum” ?

line 633: “This result poses another”

line 640: “faster algorithms for listing [or: enumerating] the” ?

Author Response

Point 1: Suggestions (include various typo corrections; better double-check for more).

Response 1: Thanks you for your suggestions. We have checked all of them and more. Furthermore, we want to highlight three particular suggestions.

Suggestion 1.1: line 258ff, line 318ff: I am unfamiliar with this kind of density estimation and the interpretation of Figures 2 and 6. What is on the x-axis? What use is the density estimation? (How) does it relate to the suggestion “it might be possible to get good representations from the distribution of weights and quotas in the target representations” in the Introduction? I would think it worthwhile to explain and motivate the allegedly “standard probabilistic approach” (line 256) to readers having a background that is different from the authors.

Response 1.1: We have detailed this point. In particular, the values on the x-axis correspond to a normalization of the considered Gaussian kernel density estimation.

Suggestion 1.2: The proofs of Prop. 4, Prop. 5 and Corollary 1 all contain too many typos for my taste. Please double-check them carefully. For instance: line 344f “However, all”; line 344 “become”; line 346 “From Proposition 1.1” (isn’t it?); line 348 “that there exists” etc. etc.

Response 1.2: We have checked all proofs.

Suggestion 1.3: line 588: It is not clear to me which notion of “statistical significance” is employed or referred to here. (I had the same problem already in line 78 but thought this would become clear later, which it didn’t.) What is the hypothesis and which test is involved? What is the relevant test statistic and which level of confidence is considered? Please either clarify or drop this point.

Response 1.3: Now, we emphasize that "Although the distributions and approximate densities might not be significant because the number of players is small, our results hint towards some promising probability distributions.".